# HNF4A and HNF1A exhibit tissue specific target gene regulation in pancreatic beta cells and hepatocytes

Natasha Hui Jin Ng [1], Soumita Ghosh[2], Chek Mei Bok[1], Carmen Ching[1], Blaise Su Jun Low [1], Juin Ting Chen[1,2,3], Euodia Lim[1,3], María Clara Miserendino[4,5], Yaw Sing Tan [5], Shawn Hoon[6] & Adrian Kee Keong Teo [1,2,3,7] ✉

*HNF4A* and *HNF1A* encode transcription factors that are important for the development and function of the pancreas and liver. Mutations in both genes have been directly linked to Maturity Onset Diabetes of the Young (MODY) and type 2 diabetes (T2D) risk. To better define the pleiotropic gene regulatory roles of HNF4A and HNF1A, we generated a comprehensive genome-wide map of their binding targets in pancreatic and hepatic cells using ChIP-Seq. HNF4A was found to bind and regulate known (*ACY3, HAAO, HNF1A, MAP3K11*) and previously unidentified (*ABCD3, CDKN2AIP, USH1C, VIL1*) loci in a tissue-dependent manner. Functional follow-up highlighted a potential role for *HAAO* and *USH1C* as regulators of beta cell function. Unlike the loss-of-function HNF4A/MODY1 variant I271fs, the T2D-associated HNF4A variant (rs1800961) was found to activate *AKAP1, GAD2* and *HOPX* gene expression, potentially due to changes in DNA-binding affinity. We also found HNF1A to bind to and regulate *GPR39* expression in beta cells. Overall, our studies provide a rich resource for uncovering downstream molecular targets of HNF4A and HNF1A that may contribute to beta cell or hepatic cell (dys)function, and set up a framework for gene discovery and functional validation.

The hepatocyte nuclear factor (HNF) family represents a group of transcription factors that play diverse but important roles in tissue development and maturation, function, and cellular metabolism. Of note, mutations in two members of this family, *HNF4A* and *HNF1A*, are known to cause Maturity Onset Diabetes of the Young (MODY) – MODY1 and MODY3 respectively[1–3]. MODY is a rare condition characterized by early age of onset and autosomal dominant inheritance of mutations in up to 14 identified genes to date. Many of these implicate pancreatic developmental genes including *HNF4A, HNF1A,* and *HNF1B*, or genes critical for pancreatic beta cell function such as *GCK* and *INS*. In the case of MODY1 and MODY3, both conditions have overlapping clinical manifestations that include defective insulin secretion capacity[4,5]. More importantly, genetic variants in both *HNF4A* and *HNF1A* have also been associated with the risk of developing the more common form of diabetes, type 2 diabetes (T2D), based on genetic and molecular evidence supporting the presence of coding variants that

[1]Stem Cells and Diabetes Laboratory, Institute of Molecular and Cell Biology (IMCB), Agency for Science, Technology and Research (A*STAR), Singapore 138673, Singapore. [2]Department of Medicine, Yong Loo Lin School of Medicine, National University of Singapore, Singapore 119228, Singapore. [3]Department of Biochemistry, National University of Singapore, Singapore 117596, Singapore. [4]Facultad de Ciencias Químicas, Universidad Nacional de Córdoba, X5000HUA Córdoba, Argentina. [5]Bioinformatics Institute, A*STAR, Singapore 138671, Singapore. [6]Molecular Engineering Laboratory, IMCB, A*STAR, Singapore 138673, Singapore. [7]Precision Medicine Translational Research Programme (TRP), National University of Singapore, Singapore 119228, Singapore. ✉e-mail: ateo@imcb.a-star.edu.sg

alter gene function to influence diabetes risk[6–8]. Therefore, a clear understanding of the gene targets and mechanisms through which HNF4A and HNF1A determine the identity and function of key metabolic tissues (such as the pancreatic beta cells and the liver) will be of immense value.

In rodents, *Hnf4a*[-/-] mice are embryonic lethal[9], but beta cell-specific *Hnf4a*[-/-] mice possess impaired glucose-stimulated insulin secretion (GSIS) leading to glucose intolerance[10,11]. *Hnf1a*[-/-] mice exhibit stunted growth and defective insulin secretory function, giving rise to abnormal glucose and development of diabetes[12,13]. More recently, human stem cell models have been used to study MODY1/MODY3 in the human context. MODY1 and MODY3 patient-derived human induced pluripotent stem cells (hiPSCs) differentiated towards the pancreatic endocrine lineage were found to exhibit downregulation of foregut genes and several pancreatic and beta cell development markers, as well as changes in the expression of beta cell stress and cellular respiration genes[14–16]. A proteomics-based pathway analysis of hPSC-derived pancreatic progenitors that underwent in vivo maturation highlighted HNF4A and HNF1A as key upstream transcriptional regulators required for commitment to the endocrine program and islet cell identity[17]. Though it is evident that HNF4A and HNF1A are key in regulating beta cell transcriptional networks during development, only Low et al. demonstrated evidence of direct gene regulation by HNF1A based on a combination of transcriptomic profiling and ChIP-Seq analyses[16].

Both HNF4A and HNF1A are involved in overlapping pathways and complex cross-regulatory feedback loops that govern both pancreas and liver development and function[18]. While an early chromatin immunoprecipitation (ChIP) study combined with promoter microarray (ChIP-on-chip) had identified HNF4A and HNF1A targets in primary human islets and primary hepatocytes[19], the downstream targets were not validated or followed up, leaving little known about the precise roles and functions of the HNF4A and HNF1A targets in pancreatic or hepatic cells. A previous HNF4A ChIP-Seq-based analysis performed on hiPSC-derived hepatic progenitors identified genes that are bound and regulated by HNF4A, that govern the transition from endoderm to hepatic cell fate[20]. However, similar high throughput data in developing pancreatic beta cells involving HNF4A or HNF1A are currently lacking.

In this resource, we have defined and provided a comprehensive dataset capturing global downstream targets of HNF4A and HNF1A based on ChIP-Seq analyses in hPSC-derived pancreatic cells, the human pancreatic beta cell line EndoC-βH1, primary human islets, hPSC-derived hepatic cells, and human hepatoma cell line HepG2. The hPSC-derived cells provided a model for capturing human cells during pancreatic and hepatic developmental time points at which HNF4A and HNF1A are most highly expressed. Biological processes and pathways that were enriched based on these HNF4A and HNF1A gene targets were investigated, and targets of higher confidence that are potentially involved in beta cell function were followed up in vitro. We further evaluated the molecular impact of the lead causal *HNF4A* T2D risk coding variant rs1800961 in human beta cells[6,7], in an attempt to uncover additional factors that may contribute to diabetes predisposition. ChIP-Seq analyses on *HNF4A* rs1800961 identified potential gene targets that could be upregulated, suggesting a gain rather than loss of function. Overall, our work provides a valuable resource of HNF4A and HNF1A targets in human pancreatic and hepatic cells, with a focus on the beta cells which lie at the heart of diabetes pathophysiology, revealing potential therapeutic targets and pathways to guide future strategies for treating diabetes.

## Results
### ChIP-Seq identifies genome-wide targets of HNF4A and HNF1A in pancreatic beta cells and hepatic cells
To identify downstream targets that are bound (and therefore potentially regulated) by HNF4A and HNF1A, we performed ChIP-Seq on a range of cell models (Fig. 1a) using a ChIP protocol optimized for both adherent cells and suspension cell clusters[21]. For hPSC-derived pancreatic cells, we selected time points at which *HNF4A* or *HNF1A* are most abundantly expressed−day 14 pancreatic progenitors (D14 PPs), day 20 endocrine progenitors (D20 EPs) and day 35 beta-like cells (D35 βLCs) for *HNF4A*[14]; D20 EPs for *HNF1A*[16]. In the context of hepatic differentiation, *HNF4A* is most highly expressed in hepatoblasts (D8 Hep)[14]. We confirmed that our selected HNF4A and HNF1A antibodies could bind specifically to their respective protein targets using immunofluorescence, western blot, and flow cytometry analyses (Fig. 1b, Supplementary Fig. 1A–D). As it is well-established that HNF4A and HNF1A exhibit promoter cross-regulation, we further confirmed that HNF4A binds on the *HNF1A* promoter (Fig. 1c) and that HNF1A binds on the *HNF4A* P2 promoter (Fig. 1d) in our ChIP samples.

Using ChIP-Seq analysis (Supplementary Fig. 1E), we assigned HNF4A- and HNF1A-bound regions to putative gene targets across the different cell types, by filtering ChIP-Seq peaks based on a false discovery rate (FDR) significance cut-off of $q = 0.1$ and mapping peaks to the nearest genes (Fig. 1e, Data S1, S2). Indeed, the top motif found for all datasets mapped to the consensus HNF4A or HNF1A motifs (Fig. 1e). Peak profiles also showed enrichment at or near to the transcription start sites (TSS) of the nearest genes (Fig. 1f). Assessment of the global genomic features showed that the percentage of promoters (up to 3 kb away from the TSS) identified across all samples ranged from ~25–70% for HNF4A, and ~15–25% for HNF1A (Fig. 1f), consistent with the expected distribution for transcription factors and previously published HNF4A/HNF1A ChIP-Seq data in liver cells from ENCODE[22]. In general, the pancreatic cell samples reported less peak enrichment regions than that observed in hepatic cells. For HNF4A ChIP-Seq, the pancreatic/endocrine progenitor samples reported less peaks than the more mature stages (Fig. 1e). These could be indicative of the size of the regulatory networks governed by HNF4A and HNF1A in the different cell types/stages.

Using a previous ChIP-on-chip dataset (in which only promoters within 1 kb were probed), in human islets for cross-reference[19], we found 88 unique gene targets of HNF4A in common (Supplementary Fig. 2A and Data S1) and 11 unique gene targets of HNF1A in common (Supplementary Fig. 2B and Data S2) with our own human islet ChIP-Seq data. The small degree of overlap in both datasets is most likely attributed to the limited genomic regions represented on the microarray[19], variability in human islet samples and differences in antibodies used. Given the lack of HNF4A or HNF1A ChIP-Seq datasets in human islets, we therefore report the most recent HNF4A- and HNF1A-bound genome-wide ChIP targets in human islets to date.

As HNF4A is also a central regulator of hepatocyte differentiation, we set out to define HNF4A targets in hepatic cells in two different cell models, (1) during early hepatic differentiation from hPSCs (in D8 Hep) when HNF4A is highly expressed, and (2) at the hepatocyte stage (in HepG2 cells). The number of ChIP peaks identified in the D8 hepatoblasts ($n = 16,697$) is far greater than that observed in HepG2 ($n = 2824$), which represents mature hepatocytes (Fig. 1e). This may be expected given that *HNF4A* expression peaks during the early progenitor stage and is decreased at the end in the hepatocyte-like cells[14], suggesting that HNF4A plays a major role in governing the transcriptional network early on during hepatic development. In comparison to a previous ChIP-Seq study of HNF4A in hiPSC-derived hepatic progenitor cells[20], we identified a subset of common gene targets in our D8 Hep ChIP-Seq dataset that may play a role in early hepatic specification (Supplementary Fig. 2C and Data S1). Several of these included *APOA2, APOB, F7, N4BP2L1, SFRP5* and *SLC35D1*, genes which were highlighted by the authors to be responsible for hepatic cell fate transition[20].

We also compared our HepG2 data with publicly-available data from ENCODE in the same cell line[22], and found that majority of our gene targets were also reported in ENCODE data (Supplementary

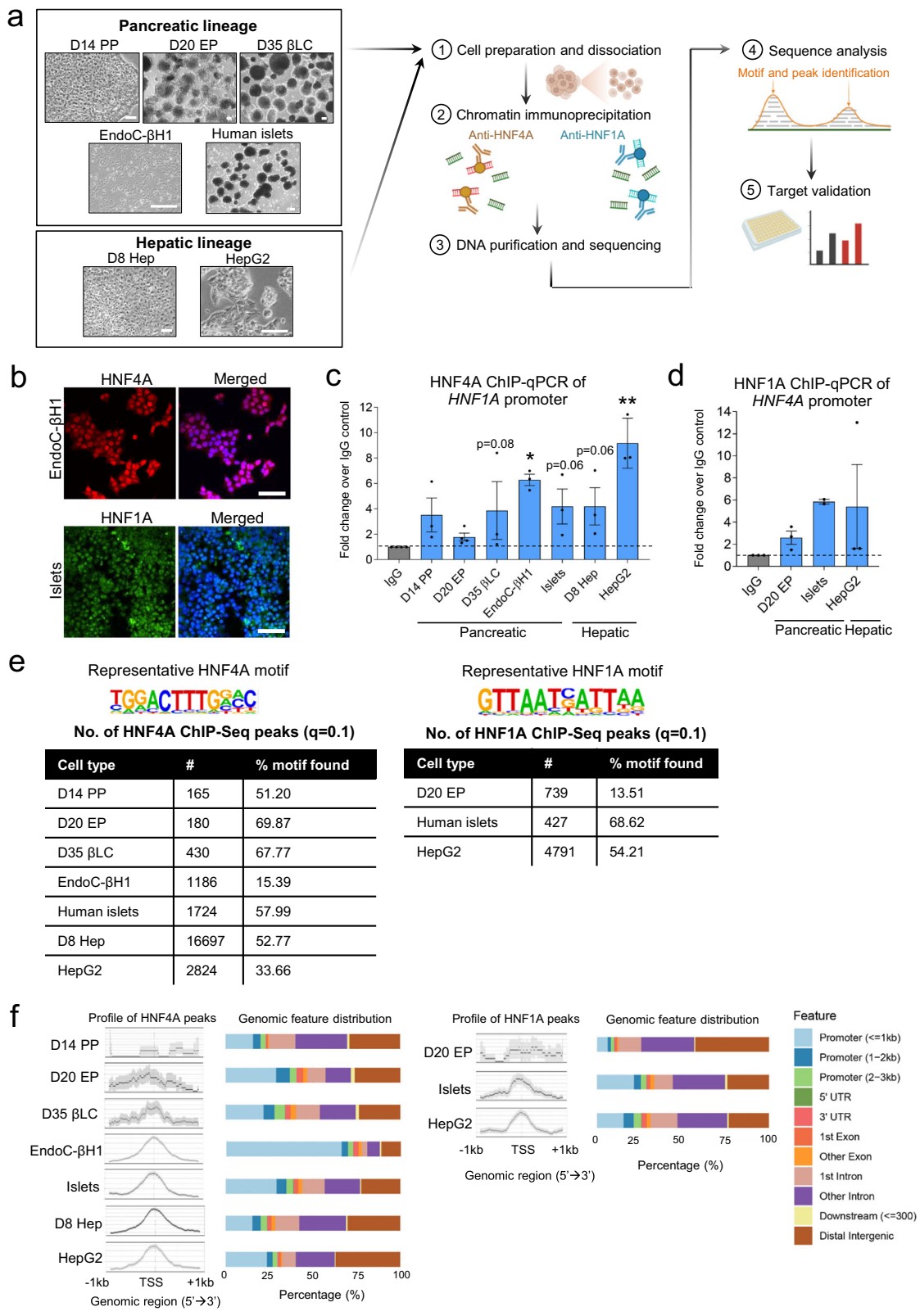

Fig. 2D and Data S1). However, as our gene list only captured a fraction of the total number of reported HNF4A-bound targets in ENCODE, we collected a replication set of HepG2 ChIP-Seq data using a different antibody (ab41898) to expand our HepG2 target list (Data S1). When combined with this replication dataset, we now capture approximately half of the unique HNF4A-bound gene targets reported in ENCODE (Supplementary Fig. 2E and Data S1). Concurrently, our expanded HepG2 dataset also reported several times more unique gene targets ($n = 4895$) than observed in the early ChIP-on-chip data in primary hepatocytes ($n = 1509$)[19] (Supplementary Fig. 2F and Data S1). As for our HNF1A ChIP-Seq targets in HepG2, we reported that majority of the unique gene targets we identified were also found within ENCODE

**Fig. 1 | Comprehensive ChIP-Seq identifies HNF4A and HNF1A targets in pancreatic and hepatic cells. a** Schematic of ChIP-Seq pipeline using multiple human cell models to identify HNF4A- and HNF1A-bound targets in pancreatic and hepatic cells. Scale bar indicates 100 μm. **b** Immunofluorescence microscopy images showing HNF4A and HNF1A nuclear localization in beta cells using the antibodies for ChIP of HNF4A (R&D Systems, H1415) and HNF1A (ab96777). Scale bar indicates 50 μm. **c** HNF4A ChIP-qPCR fold enrichment at the *HNF1A* promoter across different cell types (*n* = 4 for D20 EP; *n* = 3 for all other tissues). **d** HNF1A ChIP-qPCR fold enrichment at the *HNF4A* P2 promoter across different cell types (*n* = 2 for islets; *n* = 3 for all other tissues). **e** Representative HNF4A or HNF1A motifs identified in the ChIP-Seq data and the total number of ChIP-Seq peaks identified based on *q* value cut-off of 0.1. **f** Peak count frequency profile of HNF4A (left) or HNF1A (right) ChIP-Seq peaks that map within 1 kb of the transcription start site (TSS) across multiple cell types and the corresponding genomic feature distributions mapped to the peaks. PP Pancreatic progenitors, EP Endocrine progenitors, βLC Beta-like cells, Hep Hepatoblasts. Data are presented as mean ± SEM. Each data point represents one independent experiment. * indicates *p* < 0.05, ** indicates *p* < 0.01, relative to IgG control, using one-way ANOVA with Fisher's LSD post-hoc test. Source data and exact *P* values are provided in the Source Data file.

data[22] (Supplementary Fig. 2G and Data S2), and we identified more HNF1A-bound gene targets in HepG2 cells (*n* = 3206) than that reported in a previous ChIP-on-chip study of primary hepatocytes (*n* = 214)[19] (Supplementary Fig. 2H and Data S2).

## HNF4A-bound targets regulate overlapping and tissue-specific processes in beta cells and hepatic cells

We then asked if the identification of the downstream target genes can inform us on the cellular processes and pathways regulated by HNF4A in human beta cells and hepatic cells. Among the topmost significantly enriched Gene Ontology (GO) biological processes (BP) in EndoC-βH1 and human islets, we identified several related to cell junction assembly, cell morphogenesis, actin filament-based processes as well as protein kinase signaling (Fig. 2a and Data S3). In the hPSC-derived D35 βLCs (which presented fewer GO BP terms due to a relatively smaller number of ChIP-Seq targets identified), we similarly observed processes related to stress-activated protein kinase signaling, in addition to reactive oxygen species (ROS) metabolic processes (Fig. 2b and Data S3). The production of ROS and regulation of ROS levels in beta cells are highly relevant to its glucose-sensing function, as glycolytic flux is tightly linked with mitochondrial oxidative activity. Maintaining the balance between ROS signaling and adaptive responses to oxidative stress is necessary for normal beta cell function[23]. We also found an overrepresentation of GO BP terms related to actin filament-based processes in HepG2 cells, which represent mature hepatocytes. Thereafter, a side-by-side comparison between the beta cell line EndoC-βH1 and HepG2 cells showed that while actin filament organization and regulation of GTPase activity are processes that were commonly enriched in both cell types, other processes such as cell junction assembly, synaptic signaling and cell morphogenesis were most enriched in beta cells, whereas amine catabolic processes and regulation of peptidase activity were uniquely enriched in the hepatocytes (Fig. 2c and Data S3). A comparison of KEGG pathways across both cell types also showed enrichment in insulin secretion and cAMP signaling in EndoC-βH1 cells, in contrast with insulin/PI3K-Akt signaling pathways in HepG2 cells, consistent with differences in the established functions of these two tissue types in relation to glucose homeostasis (Supplementary Fig. 3). These findings suggest that HNF4A downstream targets may commonly influence cytoskeletal organization in both beta cells and hepatic cells and in addition, play distinct roles in the establishment of beta cell versus hepatic tissue identity and function.

As HNF4A is known to be involved in pancreatic beta cell development and function, and defects in HNF4A function can lead to diabetes, we next asked what gene targets downstream of HNF4A contribute to the regulation of beta cell function. To address this, we overlapped the HNF4A ChIP-Seq genome-wide gene targets identified in EndoC-βH1 cells, human islets, and hPSC-derived D35 βLCs, to derive a total of 63 common beta cell targets (Fig. 2d). To further narrow down to a credible set of genes that are more likely to be transcriptionally regulated by HNF4A (as they map closer to the promoter region) and therefore functionally relevant, we considered only ChIP-Seq peaks that mapped within 10 kb up and downstream of the nearest TSS, and eventually filtered down to 44 beta cell targets that we considered of high confidence (Fig. 2d and Data S4). Among these 44 genes, 28 were also found to be bound by HNF4A in the D20 EPs, indicating a potential relevance of these genes to the HNF4A regulatory network early on during endocrine development. The list of HNF4A beta cell targets included several genes previously reported (but not validated) in the ChIP-on-chip study in human islets[19], such as *ACY3*, *HAAO*, *HNF1A*, and *MAP3K11* (Data S4). *MAP3K11* belongs to the group of genes involved in the stress-activated protein kinase signaling cascade process that is found to be enriched in the beta cells (Fig. 2a, b and Data S3). Additionally, we identified HNF4A targets that appear to be involved in actin filament organization such as *USH1C* and *VIL1* (Fig. 2a, d and Data S3). These findings highlight previously uncharacterized candidate genes for further follow-up studies.

## *HAAO* and *USH1C* are among several direct targets of HNF4A that potentially contribute to beta cell function

Our ChIP-Seq data so far highlighted credible HNF4A gene targets in beta cells that warrant further studies. To validate if these genes are directly bound and regulated by HNF4A, we prioritized several HNF4A-bound target regions that map at or near to the gene promoters specifically of *ACY3*, *HAAO*, *HNF1A* (a well-established target), *MAP3K11*, *USH1C* and *VIL1* (Fig. 3a and Data S4). Direct binding at the region of peak enrichment was confirmed by qPCR in the ChIP samples for EndoC-βH1 cells or human islets (Supplementary Fig. 4A). Besides *HNF1A*, these newly-identified HNF4A targets have not been previously characterized or studied in beta cells. By tracking the gene expression patterns during beta cell differentiation from hPSCs, we found that *ACY3*, *USH1C* and *VIL1* transcripts (and to a lesser extent, *HAAO*) were increased over the course of differentiation and were most highly expressed in the end-stage D35 βLCs (Supplementary Fig. 4B), suggesting that they might play a more specific and prominent role at the beta cell stage. *ACY3* encodes aminoacylase 3 which is commonly known to catalyze the hydrolysis of N-acetyl aromatic amino acids and mercapturic acids, but has little known roles in human metabolic tissues[24]. *USH1C* encodes harmonin, a scaffold protein that is part of a protein complex involved in mechanotransduction in cochlear hair cells[25], whereas *VIL1* encodes villin-1, a Ca²⁺-regulated actin-modifying protein that has largely been studied in intestinal and kidney epithelial cells[26]. *MAP3K11* encodes a member of the serine/threonine kinase family and is a positive regulator of JNK signaling.

We next asked whether HNF4A binding leads to gene regulation. We performed an siRNA-mediated knockdown of *HNF4A* in EndoC-βH1 cells and confirmed that the gene expression levels of these targets were consequently downregulated (Fig. 3b). Conversely, we carried out transient overexpression of HNF4A to check for reverse effects. In EndoC-βH1 cells, we found that HNF4A WT significantly upregulated the expression of *ACY3*, *MAP3K11* and *VIL1*. The gene expression levels of the other targets *HAAO*, *HNF1A* and *USH1C* were also increased though the effects were modest (Fig. 3c). The T2D risk variant in HNF4A, rs1800961[7], was also able to activate gene expression similar to WT, whereas the MODY1 variant I271fs (that is known to exhibit loss-of-function [LOF])[14] displayed loss of activity and was unable to upregulate target gene expression (Fig. 3c). These

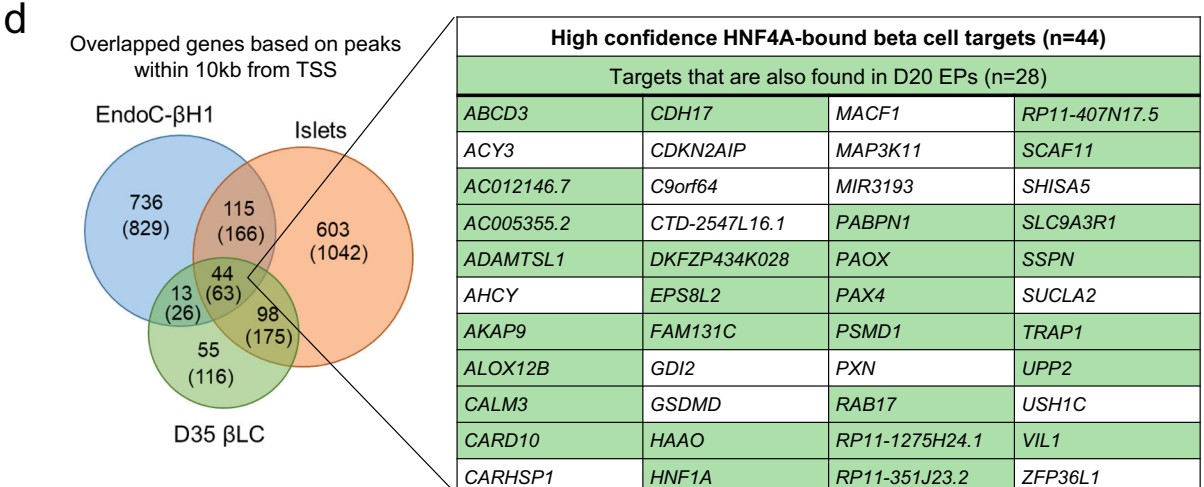

**Fig. 2 | Identification of HNF4A-bound targets in beta cells and hepatic cells reveal enrichment for common and distinct pathways across tissue types and identified beta cell target genes.** **a** Top Gene Ontology (GO) Biological Processes (BP) commonly identified in both EndoC-βH1 and human islet HNF4A ChIP-Seq. **b** Top GO BP identified in D35 βLC HNF4A ChIP-Seq, in side-by-side comparisons with EndoC-βH1 and human islet samples. **c** Top GO BP in EndoC-βH1 in comparison with HepG2 HNF4A ChIP-Seq. **d** Venn diagram showing overlaps in HNF4A-bound target genes in EndoC-βH1 cells, human islets, and D35 βLCs based on ChIP-Seq peaks mapping within 10 kb of the transcription start site (TSS) (number of total peaks with no filtering shown in brackets). The table provides a list of the common beta-cell target gene loci identified, of which some were also replicated in D20 EPs (in green). Analysis and visualization of GO data is based on the ChIPseeker R package (see Methods).

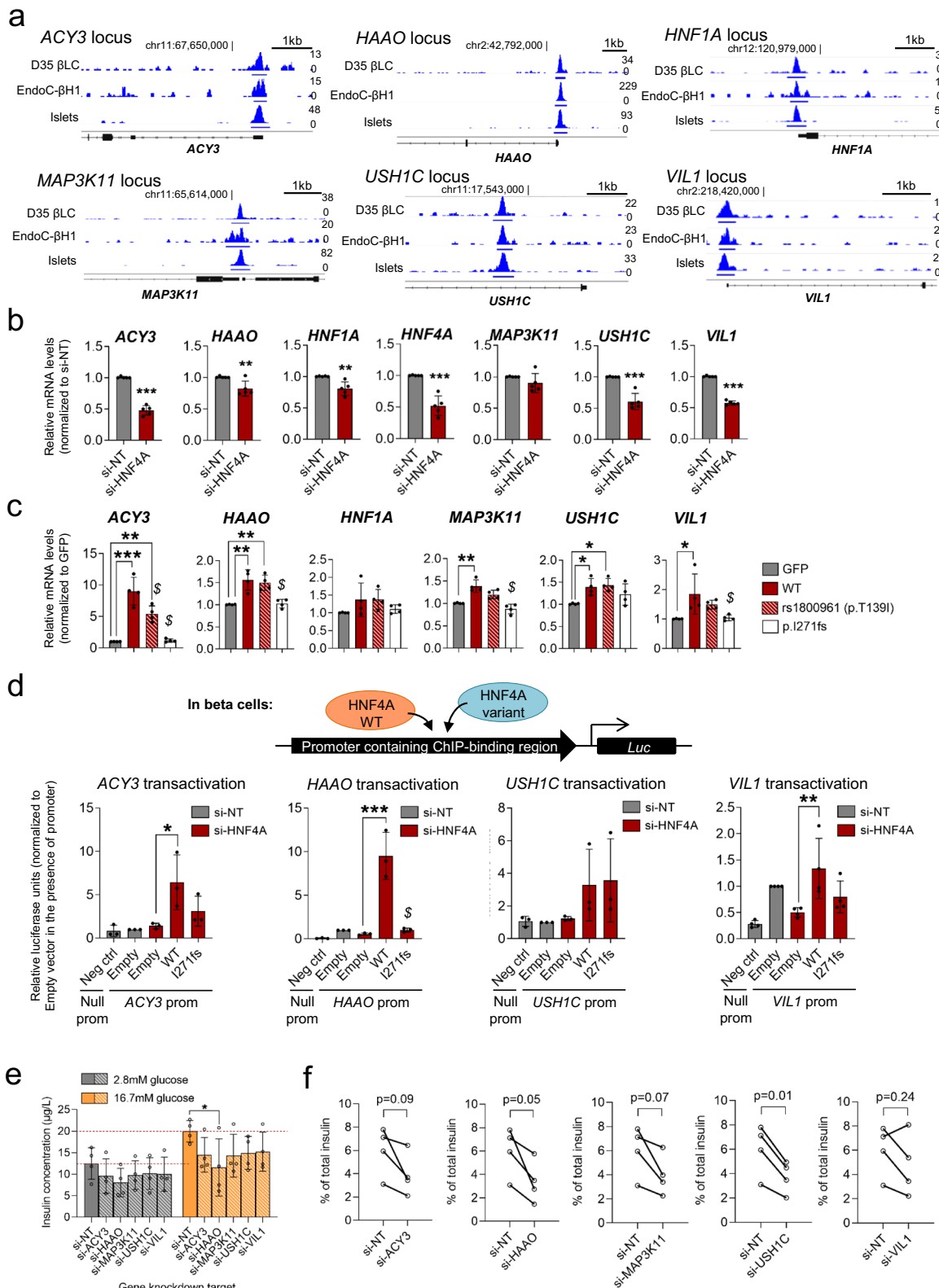

results were also replicated in Ad293 cells, a cell model that expresses low levels of endogenous HNF4A, confirming positive regulation of the tested target genes by HNF4A (Supplementary Fig. 4C).

We next set out to evaluate the direct transcriptional regulation of these gene promoters using luciferase reporter constructs containing the HNF4A-bound genomic regions. We found that the transcriptional activities at *ACY3*, *HAAO* and *VIL1* promoters were reduced after siRNA-mediated knockdown of *HNF4A* in EndoC-βH1, and subsequently rescued by the overexpression of HNF4A WT but not the LOF variant I271fs (Fig. 3d). *USH1C* promoter activity was not altered upon *HNF4A* knockdown, and HNF4A WT only modestly transactivated the *USH1C* promoter (Fig. 3d). Successful transactivation for the gene promoters at our selected loci was also

**Fig. 3 | Prioritization and functional validation of HNF4A-bound beta cell targets. a** IGV tracks showing HNF4A ChIP-Seq peaks in selected loci in D35 βLCs, EndoC-βH1 cells, and human islets. The scale used to visualize peaks in IGV is indicated on the right side of each track. The chromosomal location near the peak region is indicated. **b** Gene expression analysis of selected target genes in EndoC-βH1 cells with *HNF4A* siRNA-mediated knockdown ($n = 5$). * indicates $p < 0.05$, ** indicates $p < 0.01$, *** indicates $p < 0.001$ relative to si-NT control, based on unpaired two-tailed Students' *t* test. **c** Gene expression analysis of selected target genes in EndoC-βH1 cells overexpressing GFP (empty vector), HNF4A WT, T139I variant, or MODY1 I271fs variant constructs ($n = 4$). * indicates $p < 0.05$, ** indicates $p < 0.01$, *** indicates $p < 0.001$ relative to GFP control; $ indicates $p < 0.05$ relative to WT, based on one-way ANOVA with Tukey's post-hoc test. **d** Transactivation activities at selected target promoters in EndoC-βH1 cells using luciferase reporter assays ($n = 3$ for *ACY3/HAAO/USH1C*; $n = 4$ for *VIL1*). * indicates $p < 0.05$, ** indicates $p < 0.01$, *** indicates $p < 0.001$ relative to Empty si-HNF4A control; $ indicates $p < 0.01$ relative to WT, based on one-way ANOVA with Tukey's post-hoc test. **e** Glucose-stimulated insulin secretion (GSIS) in EndoC-βH1 cells with siRNA-mediated KD of selected target genes ($n = 4$). * indicates $p < 0.05$ relative to si-NT control, based on two-way ANOVA with Dunnett's multiple comparisons test. **f** Insulin secreted as a percentage of total insulin in EndoC-βH1 cells with siRNA-mediated KD of selected target genes, at 16.7 mM glucose stimulation ($n = 4$). *P* value indicated is based on paired two-tailed Students' *t* test. Data are presented as mean ± SD. Each data point represents one independent experiment. Source data and exact *P* values are provided in the Source Data file.

confirmed in Ad293 cells (Supplementary Fig. 4D). Though our data in Ad293 cells support the direct regulation of these genes by HNF4A, we recognize that it remains important to demonstrate this in the relevant pancreas/liver biological context.

Finally, we sought to determine if our prioritized HNF4A gene targets play a role in the regulation of beta cell function. We carried out siRNA-mediated knockdown of the five selected gene targets, namely *ACY3*, *HAAO*, *MAP3K11*, *USH1C* and *VIL1*, in EndoC-βH1 cells (Supplementary Fig. 4E), and evaluated beta cell insulin secretion. We found that knockdown of *HAAO* resulted in significant downregulation in the amount of insulin secreted in the presence of high glucose stimulation (Fig. 3e), though stimulation index remained unchanged (Supplementary Fig. 4F). When assessed as a proportion of total insulin content in the cells, knockdown of *HAAO* and *USH1C* both resulted in a significant reduction in percentage of secreted insulin (Fig. 3f). The other genes tested, *ACY3*, *MAP3K11*, *VIL1*, also displayed a downward trend in terms of percentage of insulin secreted under high glucose conditions, though the effects were not statistically significant (Fig. 3f). In all cases, total insulin content was unaffected (Supplementary Fig. 4F). *HAAO* encodes a dioxygenase that is known to catalyze the synthesis of quinolinic acid. For both *HAAO* and *USH1C*, there is no previous knowledge of their potential role in beta cells. Our findings suggest they have roles to play in regulating GSIS, that is not due to alterations in insulin production, underscoring the need for more mechanistic studies in future.

### HNF4A downstream targets signal transitions during hepatocyte maturation and pinpoint *HAAO* as a common target in both beta cells and hepatic cells

Following the pursuit of HNF4A targets in beta cells, we next asked if identification of HNF4A-bound targets in early (D8 Hep) and late hepatic cells (HepG2) can provide complementary insights into the role of HNF4A during hepatic development. In both D8 Hep and HepG2, biological processes such as axon guidance, regulation of GTPase activity and actin filament organization showed strong enrichment (Fig. 4a, Supplementary Fig. 5A and Data S3), as previously observed in the beta cells (Fig. 2 and Data S3). However, in the topmost enriched GO BP terms in D8 Hep, we also found many processes related to tissue development, synapse organization and synaptic signaling, whereas in HepG2, we found enrichment of catabolic processes, regulation of peptidase activity and carbohydrate transmembrane transport (Fig. 4a, Supplementary Fig. 5A and Data S3). These findings broadly indicated that HNF4A regulates a network of genes that are involved in many common processes over the course of hepatocyte development, but also sees a shift from developmental processes and intercellular signaling to catabolic functions that are in line with the vital role of the liver in energy metabolism and production.

As HNF4A binds to a large number of targets in hepatic cells, we sought to narrow down to a credible set of hepatic targets to facilitate downstream validation. We identified gene targets that were pulled down using two different HNF4A antibodies in HepG2 cells (Data S1), narrowing down to a consensus list of 876 HNF4A-bound gene targets in

HepG2 cells (Data S5). We further intersected our HepG2 data with the D8 Hep data to derive a list of 717 targets, of which 391 were considered as HNF4A-bound hepatic gene targets of high confidence as the ChIP-Seq peaks mapped nearer (within 10 kb) to the TSS (Supplementary Fig. 5B and Data S5). Some notable groups of genes that are found within these high confidence hepatic cell targets include those that are related to the fatty acid catabolic process (such as *ABCD3*, *ECH1*, *ETFA*, *LPIN3*), as well as cholesterol homeostasis (such as *APOA1*, *APOC3*, *EPHX2*, *NR5A2*) (Supplementary Fig. 5B and Data S5), revealing some common mechanisms that HNF4A is likely to be involved in throughout the course of hepatocyte development. This is also consistent with a previous study of hPSC-derived hepatic progenitors that had identified cholesterol metabolism to be essential for hepatocyte specification[27].

We next asked whether there are any robust HNF4A targets that are agnostic to tissue type and may play similar roles in both beta cells and hepatic cells. To address this, we identified the high confidence gene targets that are commonly bound by HNF4A in both beta cells (Fig. 2d) and hepatic cells (Supplementary Fig. 5B), at the same time revealing genes that are bound in a cell type-specific manner (Fig. 4b). Twenty-eight genes were bound by HNF4A in both cell types, including *ABCD3*, *CDKN2AIP*, *HAAO*, *HNF1A* (well-established target) and *MAP3K11* (Fig. 4c). In contrast, genes such as *ACY3*, *USH1C* and *VIL1* which we highlighted earlier, were not bound by HNF4A in hepatic cells, suggesting that they are likely to be beta cell-specific targets of HNF4A, at least based on our ChIP-Seq data. Using ChIP-qPCR analyses, we validated enrichment at the promoter regions of *ABCD3*, *CDKN2AIP*, *HAAO* and *MAP3K11* in hepatic cells (Supplementary Fig. 5C), and further confirmed that HNF4A WT but not the MODY1 I271fs variant significantly transactivated *CDKN2AIP*, *HAAO* and *MAP3K11* promoters based on luciferase reporter assays in HepG2 cells (Fig. 4d), consistent with what was observed in Ad293 cells (Supplementary Fig. 4D). At *ABCD3* promoter, transactivation was also observed in Ad293 cells (Supplementary Fig. 4D). Together, these results suggest that transcriptional regulation at each gene locus is cell-type dependent. The expression of the four genes evaluated here, *ABCD3*, *CDKN2AIP*, *HAAO* and *MAP3K11*, tended to peak at D8 in the hepatoblasts, and continue to be expressed throughout hPSC-based hepatic differentiation into hepatocyte-like cells (Supplementary Fig. 5D), consistent with our knowledge and findings in this study that *HNF4A* is highly expressed early during hepatic differentiation[14] and regulates the gene expression patterns of its targets in the progenitor cells.

Taken together, we have so far identified a set of common HNF4A targets in both beta cells and hepatic cells, such as *HAAO*, suggesting common role(s) in these cell types. In addition, these comparisons have helped to identify unique beta cell-specific HNF4A targets that are well-positioned for in-depth follow-up in relation to beta cell function (or dysfunction).

### Identification of HNF1A-bound targets in pancreatic endocrine cells reveals several candidate target genes

Building on our approach to identify HNF4A-bound targets, we utilized a similar pipeline to determine the genes and pathways regulated by

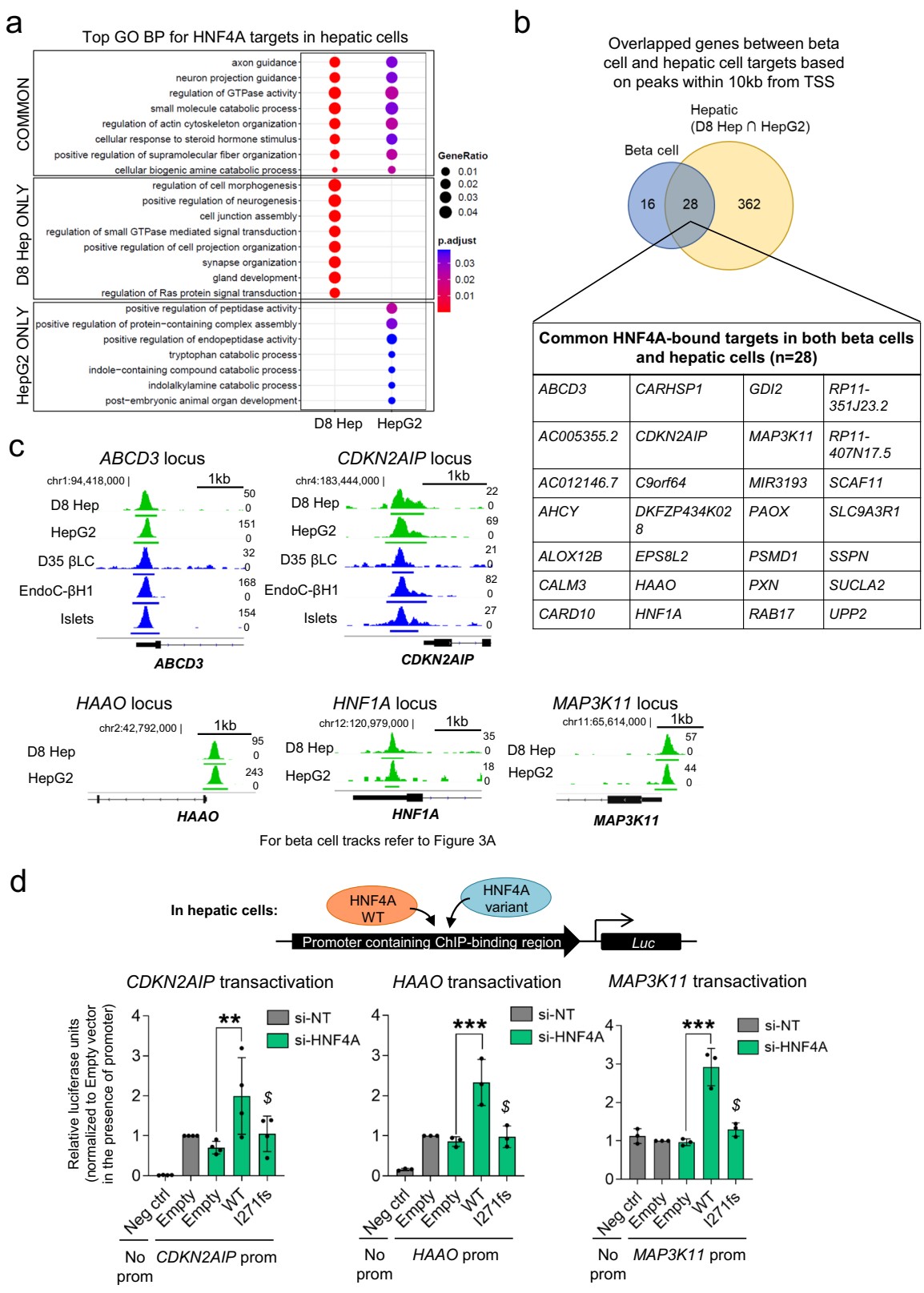

HNF1A, another transcription factor of the same family in beta cells and hepatic cells. As HNF1A ChIP-Seq in the EndoC-βH1 cells yielded less than 20 peaks and were thus not included in this study, we relied on human islet data (Data S2). We first evaluated the biological processes regulated by HNF1A-bound targets in human islets. The topmost enriched GO BP terms were related to regulation of cell adhesion

and migration, tissue development and morphogenesis (Fig. 5a and Data S6). To identify HNF1A targets in pancreatic endocrine cells, we overlapped the HNF1A-bound gene targets for D20 EPs (during which HNF1A is highly expressed) and human islets. We identified a total of 72 HNF1A beta cell targets, of which 30 were based on peak regions that mapped within 10 kb of the TSS (Fig. 5b and Data S7), and were

**Fig. 4 | HNF4A downstream targets signal changes across hepatic cells and pinpoint common targets in both beta cells and hepatic cells. a** Topmost common and distinct Gene Ontology (GO) biological processes (BP) of HNF4A ChIP-Seq targets in D8 hepatoblasts and HepG2 cells. Analysis and visualization of GO data is based on the ChIPseeker R package (see Methods). **b** Venn diagram showing overlaps in HNF4A-bound beta cell and hepatic cell target genes based on ChIP-Seq peaks within 10 kb of the transcription start site (TSS). Table provides a list of gene loci identified in both cell types. **c** IGV tracks showing ChIP-Seq peaks that map to the nearest genes in selected loci in hepatic cells. The scale used to visualize peaks in IGV is indicated on the right side of each track. The chromosomal location near the peak region is indicated. **d** Luciferase reporter analysis of *CDKN2AIP*, *HAAO* and *MAP3K11* promoter activities in HepG2 cells ($n = 3$ for *HAAO/MAP3K11*; $n = 4$ for *CDKN2AIP*). Data are presented as mean ± SD. Each data point represents one independent experiment. *** indicates $p < 0.001$, ** indicates $p < 0.01$, relative to Empty si-HNF4A control in the presence of the promoter. \$ indicates $p < 0.05$ relative to WT, based on one-way ANOVA with Tukey's post-hoc test. Source data and exact $P$ values are provided in the Source Data file.

considered high confidence beta cell targets. Further to observing peak enrichment at the known *HNF4A* promoter (though the peak did not achieve the significance cut-off of $q = 0.1$), we also identified *NR5A2*, which encodes a transcription factor involved in cholesterol homeostasis and triglyceride synthesis (Fig. 5b, c and Data S7). Other targets included *GPR39* (encodes a zinc-binding receptor that is a member of the ghrelin receptor family), which was recently shown to be a target of HNF1A[16], and *COBLL1* and *HGD* (Fig. 5b, c). As before, we confirmed peak enrichment at the respective gene loci via ChIP-qPCR in beta cell samples (Supplementary Fig. 6A). As the biological relevance of these genes in beta cells is unclear, we evaluated their expression patterns during beta cell differentiation, and found that *NR5A2* is constitutively expressed at all time points during hPSC-based beta cell differentiation (Supplementary Fig. 6B), indicating a role that is not specific to the late beta cell stage. In contrast, *COBLL1*, *GPR39* and *HGD* levels were progressively increased during differentiation into D35 βLCs (Supplementary Fig. 6B), suggesting that these genes may play more dominant roles in the regulation of beta cell function.

Following these observations, we also asked whether HNF1A-binding resulted in gene regulation. Upon knockdown of *HNF1A* in EndoC-βH1 cells via siRNA, apart from *NR5A2*, we showed that *COBLL1*, *GPR39*, *HNF4A*, and *HGD* were consequently downregulated (Fig. 5d), indicating that these are indeed functional targets of HNF1A. As a proof-of-concept, transactivation of the *GPR39* promoter by HNF1A was evaluated using a luciferase reporter assay. We showed that *GPR39* activation was downregulated upon *HNF1A* knockdown in EndoC-βH1 cells, whereas overexpression of HNF1A WT was able to rescue promoter activity (Fig. 5e). In contrast, the LOF MODY3 variant P291fsinsC was unable to activate promoter activity (Fig. 5e). These results were further replicated in transactivation assays performed in Ad293 cells, which also express low endogenous levels of HNF1A (Supplementary Fig. 6C). Though we have not performed further validation of other HNF1A targets highlighted here, they represent candidate genes that can be followed up in future studies. Overall, we have identified HNF1A targets that may play functionally important role(s) in beta cells, providing a valuable resource for further data mining.

## HNF4A and HNF1A bind to common beta cell targets
HNF4A and HNF1A are known to govern overlapping transcriptional networks and therefore share common transcriptomic signatures[28]. As both HNF4A-MODY1 and HNF1A-MODY3 are also clinically associated with defective beta cell insulin secretion function, we sought to compare HNF4A and HNF1A ChIP-Seq targets in beta cells, expecting to find overlapping genes and pathways. To do this, we compared the HNF4A and HNF1A ChIP-Seq data collected from the same human islet preparation (Data S1 and S2). A side-by-side comparison of GO BP enrichment signatures revealed that actin filament-based processes and regulation of cell adhesion and migration represented the topmost commonly enriched terms (Fig. 6a), supporting the underappreciated knowledge that cytoskeletal structure and extracellular interactions (with the external environment or between cells) are important for beta cell function[29], and that these are in part controlled by both HNF4A and HNF1A. As for distinct pathways, we found that HNF4A targets are enriched for protein kinase signaling processes (Fig. 2), whereas HNF1A targets are enriched for several tissue

development and differentiation processes (Fig. 6a), indicating some level of divergence in the regulatory roles that both transcription factors play in mature beta cells. A direct comparison between HNF4A- and HNF1A-bound targets in islet cells led to 32 genes (from peaks mapping within 10 kb of the TSS) that are common, including *EEF1A1*, *HGD*, *NR5A2*, and *PAX4* (Fig. 6b and Data S8).

Both *EEF1A1* and *NR5A2* were also commonly-bound targets by HNF4A and HNF1A in HepG2 cells, among a total of 374 such genes, of which 142 mapped close to the TSS (within 10 kb) of the respective genes (Fig. 6c and Data S8). In HepG2 cells, common pathways that were enriched included actin filament-based processes (also found in human islet data) and regulation of GTPase activity (Fig. 6d). However, the topmost enriched processes distinctly governed by HNF4A targets include catabolic processes, whereas those mediated distinctly through HNF1A targets again involve several tissue development-related terms (Fig. 6d). Taken together, our comprehensive HNF4A and HNF1A ChIP-Seq data provide the opportunity to interrogate the overlapping and distinct downstream targets mediated by both proteins in beta cells and hepatic cells, and consequently, the biological processes they are regulating.

## HNF4A T2D risk variant rs1800961 ectopically upregulates the expression of specific target genes in beta cells
Finally, we asked whether our approach can shed light on the molecular impact of the *HNF4A* T2D risk variant rs1800961. We generated stable lines overexpressing FLAG-tagged HNF4A WT or risk variant rs1800961 and showed that the variant displayed similar expression patterns as WT protein based on western blot analysis and cellular localization studies in both EndoC-βH1 (Fig. 7a, b) and Ad293 cells (Supplementary Fig. 7A). ChIP pulldown using the FLAG antibody confirmed successful enrichment at the *HNF1A* promoter region in WT- or variant-expressing EndoC-βH1 cells (Fig. 7c). We then conducted ChIP-Seq analysis to map the genome-wide binding targets of rs1800961 compared with WT, to identify potential alterations in transcriptional signatures and changes in protein function due to the single base substitution. Given that the small selection of targets we have tested so far do not appear to be dysregulated by this T2D risk variant (Fig. 3c and Supplementary Fig. 4C), we postulated that variant effects may lie in subtle differences in the gene regulation of specific downstream targets that remain to be defined.

In this case, ChIP-Seq of the canonical HNF4A isoform (HNF4A2) in EndoC-βH1 cells yielded minimal binding targets. Therefore, we focused our analysis on another full-length isoform (that differed only at the N-terminal exon) known as isoform 8 (or HNF4A8), in which rs1800961 results in p.T117I in HNF4A8 (Supplementary Fig. 7B). Though both isoforms studied are expressed in pancreatic cells[14,30], the HNF4A8 isoform is known to be driven by a different distal promoter and was earlier reported to be more abundantly expressed in pancreatic islets[31,32]. Reassuringly, the consensus HNF4A motif was identified at a frequency of 54–72% in all identified peak regions (Fig. 7d). Consistent across all WT and T117I datasets, ~25% of the ChIP-Seq peaks mapped within the promoter region and peak signals were enriched at the TSS (Fig. 7e and Supplementary Fig. 7C). In comparing the targets bound by WT and T117I, we found that binding at several HNF4A targets such as *HNF1A* and *ACY3* were indeed no different between WT

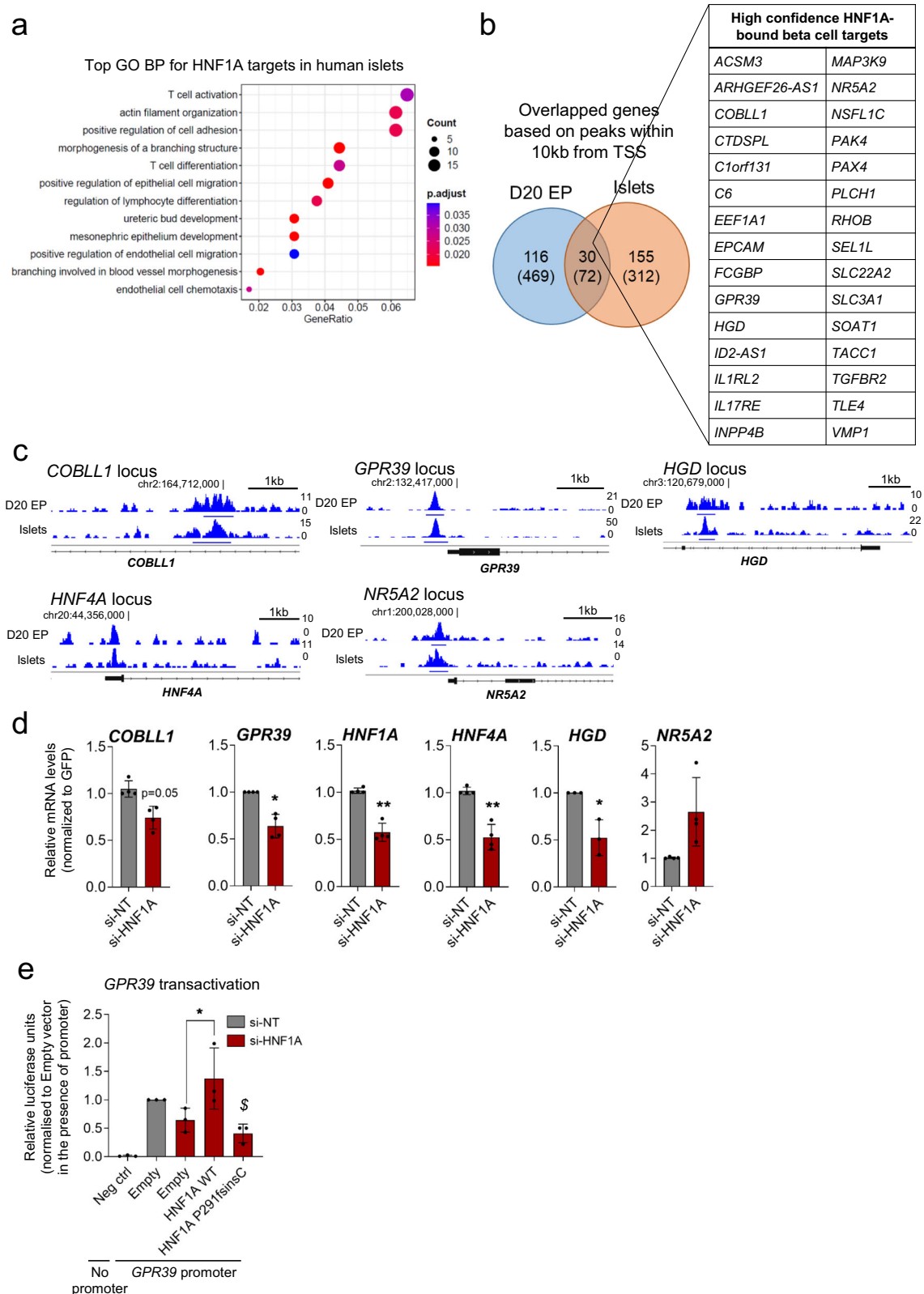

and T117I (Supplementary Fig. 7D), and were consistent with our earlier HNF4A ChIP-Seq data in EndoC-βH1 cells (Fig. 3a).

To identify differentially-bound regions, we compared the commonly enriched regions between WT and T117I, manually curated gene loci, and discovered at least three loci at which there was an enrichment of binding by T117I but not (or to a much smaller extent) by WT,

namely *AKAP1 (promoter)*, *GAD2* (intronic) and *HOPX (intronic)* (Fig. 8a). Differences in binding affinity at these sites were further confirmed using ChIP-qPCR validation (Fig. 8b). The *AKAP1*-bound region maps close to the promoter of the gene, with some evidence of peak enrichment in our earlier HNF4A ChIP-Seq data in EndoC-βH1 cells, similar to control WT cells (Supplementary Fig. 7E). In the case of

**Fig. 5 | Identification of HNF1A-bound targets in pancreatic endocrine cells highlight several target genes regulated by HNF1A. a** Topmost gene ontology (GO) biological processes (BP) found in HNF1A ChIP-Seq targets in human islets. Analysis and visualization of GO data is based on the ChIPseeker R package (see Methods). **b** Venn diagram showing overlaps in HNF1A-bound target genes in D20 EPs and human islets based on ChIP-Seq peaks within 10 kb of the transcription start site (TSS) (number of total peaks with no filtering shown in brackets). Table provides a consensus list of the beta cell target genes identified. **c** IGV tracks showing HNF1A ChIP-Seq peaks that map to the nearest genes in selected loci in D20 EPs and human islets. The scale used to visualize peaks in IGV is indicated on

the right side of each track. The chromosomal location near the peak region is indicated. **d** Gene expression analysis of selected target genes in EndoC-βH1 cells with *HNF1A* siRNA-mediated knockdown ($n = 3$). * indicates $p < 0.05$, ** indicates $p < 0.01$ relative to si-NT (non-targeting control) using paired two-tailed Students' *t* test. **e** Transactivation activity of *GPR39* promoter in EndoC-βH1 cells ($n = 3$). * indicates $p < 0.05$ relative to Empty si-HNF1A control in the presence of the *GPR39* promoter; $ indicates $p < 0.01$ relative to WT, based on one-way ANOVA with Tukey's post-hoc test. Data are presented as mean ± SD. Each data point represents one independent experiment. Source data and exact *P* values are provided in the Source Data file.

*GAD2*, the peak region mapped to an enhancer cluster found in human islet chromatin state data, and *GAD2* has been identified as an islet-specific gene[33], supporting our hypothesis that an increase in *GAD2* may have a functional consequence in beta cells. To determine if our observations translated to changes in gene expression levels, we showed in our WT- or T117I-overexpressing EndoC-βH1 cells that modest but significant upregulation of gene expression was observed for *GAD2* and *HOPX* (Fig. 8c), suggesting that T117I may exhibit ectopic gain-of-function by binding to and activating the transcription of these gene targets. We also checked if the HNF4A2 isoform displayed similar differential binding patterns, and found that there were no significant differences observed between HNF4A2 WT and the corresponding T139I variant (Supplementary Fig. 7F), likely supporting the notion that transcription regulation of specific gene targets can occur in an isoform-dependent manner[34].

To then understand the molecular basis of potential gain-of-function effect of HNF4A rs1800961, we performed molecular dynamics (MD) simulations of the complexes of WT and mutant HNF4A with DNA. The crystal structure of HNF4A bound to DNA[35] shows that the HNF4A homodimer binds in an asymmetric fashion to its DNA response element (Fig. 8d). T117 (the residue position based on the WT HNF4A8 isoform sequence will be used in this section) is located in the hinge region between the DNA binding domain and the ligand binding domain (LBD), close to the DNA in the downstream HNF4A subunit and further away from the DNA in the upstream HNF4A subunit (Fig. 8d). The simulations suggested that the mutation does not result in a significant difference in the binding free energies compared to the WT (Data S9). As HNF4A is known to be regulated by a variety of post-translational modifications[36,37], we considered the possibility of T117 being a phosphorylation site. Indeed, it has been reported that this site could be phosphorylated by ERK1/2[38]. The NetPhos 3.1 web server[39], which uses a neural network-based method to predict potential phosphorylation sites, also predicted with high confidence that T117 could be phosphorylated by either an unspecified kinase (likely ERK1/2) or protein kinase C. We investigated the effect of T117 phosphorylation (or pT117) on the DNA-binding ability of HNF4A in MD simulations. The computed binding free energies based on the trajectory structures suggested that pT117 significantly decreases the binding affinity of HNF4A for DNA (Data S9) compared to both WT and T117I mutant HNF4A. Based on the breakdown of the free energy components, the difference is mostly attributed to the electrostatic interactions. This is due to the close proximity of the downstream HNF4A subunit's phosphorylated T117 to the negatively charged DNA backbone (Fig. 8d), resulting in unfavorable electrostatic repulsion and therefore attenuating the binding affinity of phosphorylated HNF4A for DNA. As our structural modeling suggested that the T117I mutation prevents the phosphorylation-mediated downregulation of HNF4A binding to DNA, this may explain the enrichment of binding by T117I to certain gene loci compared to WT. Overall, as T2D risk variants like HNF4A rs1800961 are not expected to give rise to severe phenotypes as compared to causal mutations for MODY1, our experimental work uncovered specific changes in gene regulation caused by the variant and shed light on the potential molecular mechanism accounting for differences in DNA-binding which have not been revealed before.

## Discussion

The HNF family of transcription factors has long been known to be important regulators of tissue development and metabolism[40] especially in the pancreas, liver, kidney, and small intestines, where they are most highly expressed[18]. HNF4A and HNF1A govern transcriptional networks that are more well studied in the liver and during hepatocyte differentiation. Much of the early work was also conducted in rodent models, leaving gaps in the knowledge pertaining to their roles in human pancreas development and function. Genetic variants in *HNF4A* and *HNF1A* are known to be causal for MODY (through a beta cell defect) and have also been robustly associated with T2D risk, implicating a clear role in regulating pancreatic beta cell function. This begs the question of how both genes regulate insulin secretion and through which targets they do so. An earlier ChIP-microarray study provided limited information on the binding regions and may not have captured all possible HNF4A and HNF1A binding targets[19].

In the present study, we conducted unbiased ChIP-Seq analyses in multiple human cell models, followed by target prioritization and systematic molecular validation of selected gene targets. We focused our efforts primarily on pancreatic beta cells and secondarily on hepatic cells, as these cells are central to diabetes biology and provide a biologically relevant context for the study of HNF4A and HNF1A. We utilized hPSC-based differentiated cells to capture early developmental stages in the human context for interrogating HNF4A and HNF1A targets, which is not possible to do using terminally differentiated cell lines or primary cells. The data allowed us to make comparisons of the downstream biological processes mediated by these two functionally important transcriptional regulators in beta cells and hepatic cells, and hone in on robust or high confidence target genes that may provide insight(s) into understudied mechanisms that regulate beta cell function. A limitation of our hPSC-based models however relates to the heterogeneity of differentiated cells. For instance, the beta cell differentiation protocol used in this study generates ~35% insulin-positive cells based on previously published flow cytometry data[16,41], indicating the presence of other non-βLC populations. As ChIP-Seq was conducted on bulk samples, peak signals from the D35 βLC samples could be derived from non-insulin-expressing cells as well. To address this, we overlapped the hPSC-derived cell data with our EndoC-βH1 (a homogenous human β cell line) and human islet datasets to focus only on gene targets that are common. This increases the confidence that the genes highlighted are of relevance to β cells.

Next, through GO enrichment analyses of biological processes and KEGG pathways, we identified many pathways downstream of HNF4A that are related to actin cytoskeleton structure and organization. Previous studies have showed that actin organization and glucose-dependent F-actin remodeling is critical for secretory granule localization and insulin release from beta cells[42–44]. Modulation of actin polymerization has also been reported to regulate the efficiency of endocrine induction and consequently beta cell function during the generation of hPSC-derived βLCs[45]. Cytoskeleton dynamics is also closely tied with GTPase signaling activity, cell adhesion and migration[46], processes that were identified in our GO BP enrichment analyses in beta cells. Notably, several catabolic processes and amino

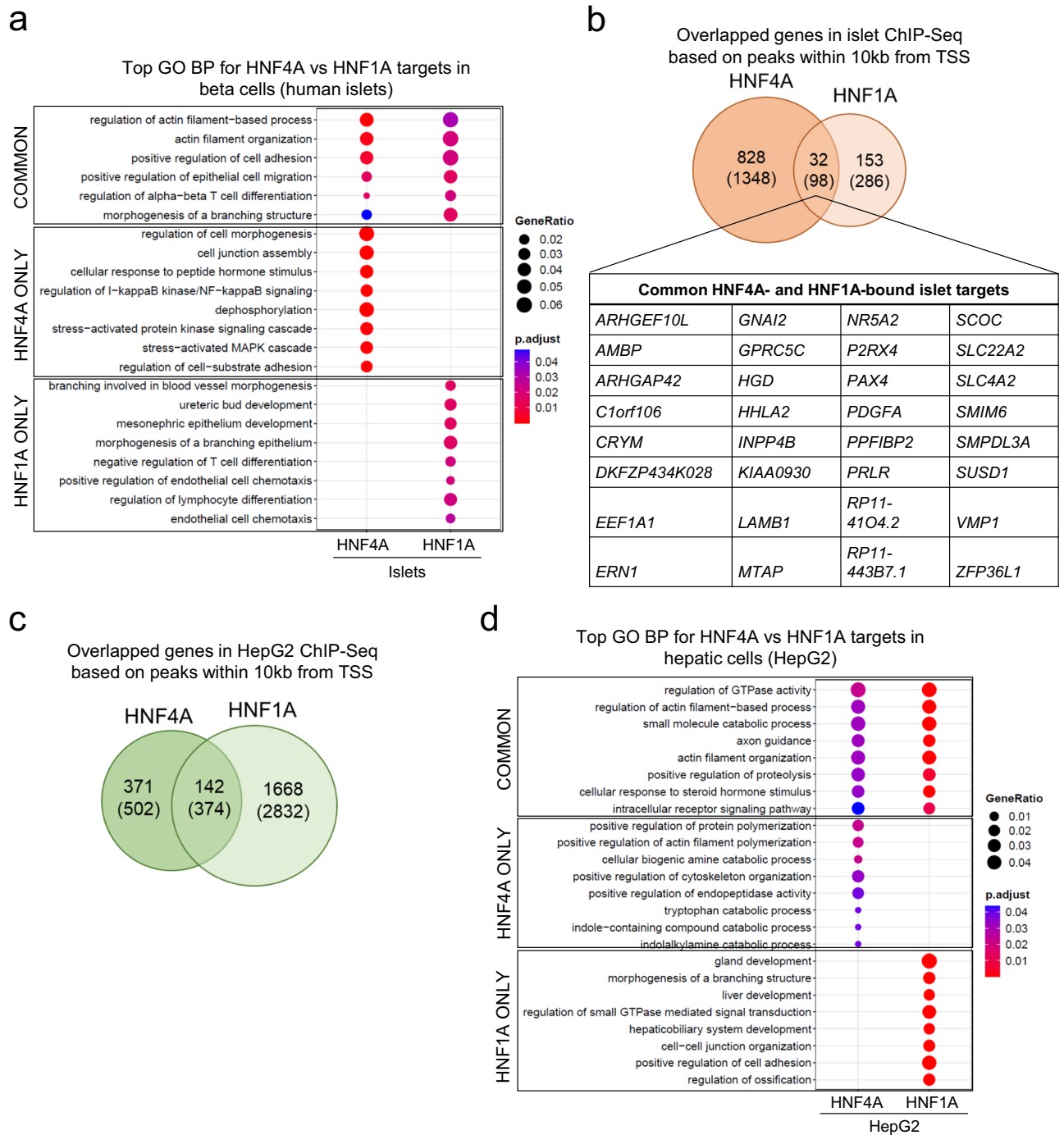

**Fig. 6 | HNF4A and HNF1A commonly bind to several beta cell targets and govern overlapping pathways in both beta cells and hepatic cells. a** Topmost common and distinct gene ontology (GO) biological processes (BP) from HNF4A- and HNF1A-bound target genes in human islets. **b** Venn diagram showing overlap in HNF4A- and HNF1A-bound target genes in human islets based on ChIP-Seq peaks within 10 kb of the transcription start site (TSS) (number of total peaks with no filtering shown in brackets). Table provides a consensus list of the commonly- bound target genes. **c** Venn diagram showing overlaps in HNF4A- and HNF1A-bound target genes in HepG2 cells based on ChIP-Seq peaks within 10 kb of the transcription start site (TSS) (number of total peaks with no filtering shown in brackets). HNF4A-bound targets in HepG2 cells are from the consensus list of genes. **d** Topmost common and distinct gene ontology (GO) biological processes (BP) from HNF4A- and HNF1A-bound target genes in HepG2 cells. Analysis and visualization of GO data is based on the ChIPseeker R package (see Methods).

acid metabolic processes were found to be more enriched in the hepatic cells than in beta cells, highlighting clear differences in the transcriptional network governed by HNF4A that is commensurate with tissue function. Nonetheless, we recognize that our approach of analyzing transcription factor-bound targets is an extrapolation of potential pathways that may be at play in the various cell types, and does not provide claim to differentially expressed genes and pathways.

Complementing our approach with differential transcriptomic data can help to address this limitation. Further studies beyond this human cell-based HNF4A and HNF1A resource that we generated are certainly warranted.

We subsequently identified and validated several targets to confirm that they are directly regulated by HNF4A or HNF1A. A number of these exhibited strong biological candidacy for a potential role in

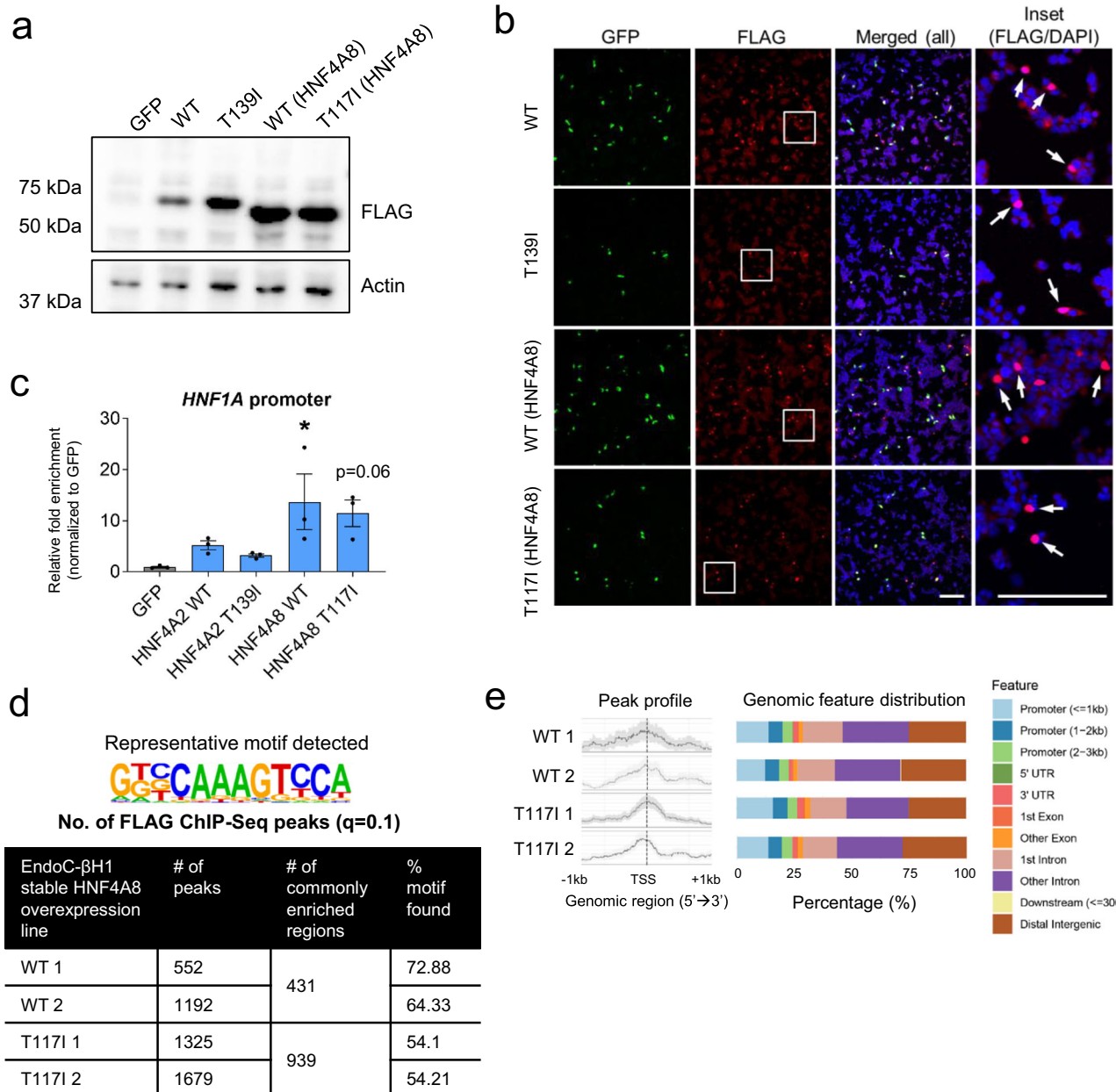

**Fig. 7 | Investigation of the effects of *HNF4A* T2D risk variant rs1800961 on gene regulation using FLAG ChIP-Seq. a** Western blot analysis of cell lysates from EndoC-βH1 cells stably overexpressing FLAG-tagged HNF4A WT and T2D variant constructs. **b** Immunofluorescence microscopy images showing nuclear localization of FLAG-tagged HNF4A WT and T2D variants overexpressed in EndoC-βH1 cells, with constitutive GFP signal from the vector in green, FLAG-tagged protein in red and DAPI in blue. Enlarged images from the insets are shown in the last panel. Scale bar indicates 100 μm. **c** HNF4A FLAG ChIP-qPCR fold enrichment at the *HNF1A* promoter in the stable EndoC-βH1 cell lines (*n* = 3). * indicates *p* < 0.05 relative to GFP control based on one-way ANOVA with Dunnett's post-hoc test. **d** Representative HNF4A motif identified in the ChIP-Seq samples and the total number of ChIP-Seq peaks identified in each sample based on *q* value cut-off of 0.1. **e** Peak count frequency profile of HNF4A FLAG ChIP-Seq peaks that map within 1 kb of the transcription start site (TSS) across the EndoC-βH1 stable lines and the corresponding genomic feature distributions. Data are presented as mean ± SD. Each data point represents one independent experiment. Source data and exact *P* values are provided in the Source Data file.

insulin secretion, such as *HAAO* and *USH1C*. *HAAO* is robustly bound by HNF4A in both beta cells and hepatic cells. It catalyzes the synthesis of quinolinic acid (QA) from 3-hydroxyanthranilic acid, which is the last step in the kynurenine pathway[47], a major tryptophan metabolic route in the cell. QA contributes to NAD+ production which is an important regulator of energy homeostasis. The tryptophan/kynurenine pathway (TKP) has been shown to be sensitive to inflammation and glucolipotoxicity, and influences GSIS in rodent islet cells[48]. QA is known to be an NMDA receptor agonist, and NMDA receptor activation appears to reduce GSIS[49]. Given the multiple roles of QA, it is yet unclear what the

precise effects of HAAO activity are in human islet cells. *USH1C* encodes harmonin, which is known to be expressed in sensory cells such as the cochlear hair cells of the inner ear to mediate mechanoelectrical transduction[25]. Mutations in *USH1C* cause Usher syndrome, a disorder that leads to hearing and vision loss[50]. Again, a link between *USH1C* and beta cell biology has not been established before. *VIL1* is another gene target that is related to the cytoskeletal compartment of cells. *VIL1* expression levels were found to be downregulated in MODY1 hiPSC-derived pancreatic progenitors in an earlier study of the transcriptional alterations due to the HNF4A MODY1 mutation[14].

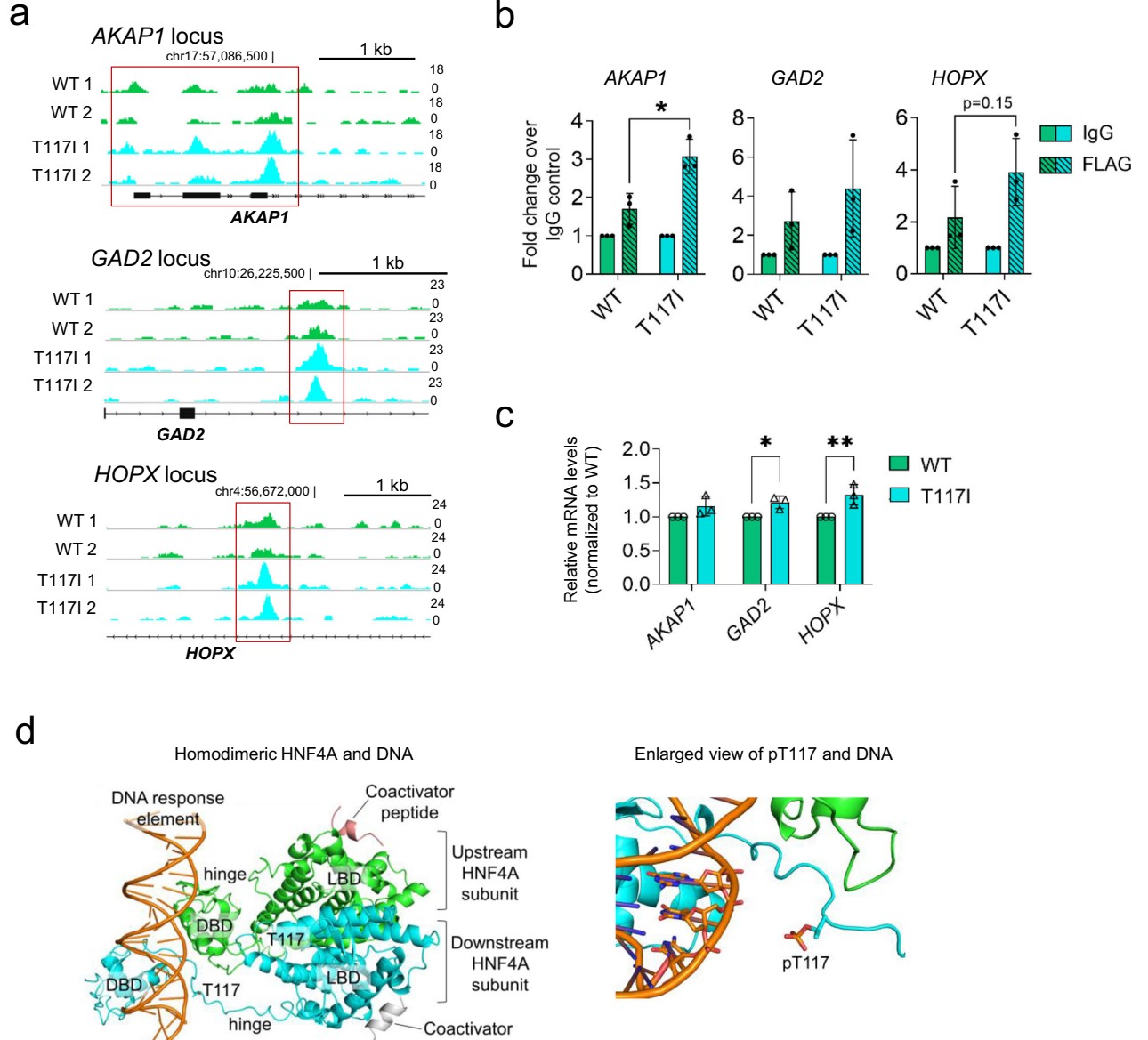

**Fig. 8 | Determining the regulation of target genes by *HNF4A* T2D risk variant rs1800961. a** IGV tracks showing FLAG ChIP-Seq peaks that map to the *AKAP1*, *GAD2*, and *HOPX* loci, in which differential binding between HNF4A8 WT and T117I was observed. Tracks within each locus have been adjusted to the same scale as specified on the right. The chromosomal location near the peak region is indicated. **b** FLAG ChIP-qPCR fold enrichment at the selected target regions in the stable EndoC-βH1 cell lines expressing HNF4A8 WT or T117I (*n* = 3). * indicates *p* < 0.05, ** indicates *p* < 0.01 based on two-way ANOVA. **c** Gene expression analyses of differentially-bound target genes *AKAP1*, *GAD2*, and *HOPX* in EndoC-βH1 cells expressing HNF4A8 WT or T117I (*n* = 3). * indicates *p* < 0.05, ** indicates *p* < 0.01

based on two-way ANOVA with Bonferroni's multiple comparisons test. **d** Structural modeling shows the crystal structure of homodimeric HNF4A (upstream subunit in green, downstream subunit in cyan) bound to its DNA response element (orange), coactivator peptides (salmon and white) and myristoate (PDB code 4IQR[35]) (left), and the final trajectory structure from a representative molecular dynamics (MD) simulation of phosphorylated HNF4A complexed with DNA (right). Phosphorylated T117 (pT117) in the hinge region of the downstream subunit and nearby DNA nucleotides are shown in sticks. Data are presented as mean ± SD. Each data point represents one independent experiment. Source data and exact *P* values are provided in the Source Data file.

As a modifier of actin filaments, it was shown in a previous report that depletion of villin in mouse islets resulted in enhanced basal mobility of the secretory granules in beta cells, resulting in increased basal insulin secretion but consequently reduced GSIS[51]. We found that siRNA-mediated knockdown of *VIL1* in human beta cells similarly result in a downward trend in GSIS. *ACY3* is another previously uncharacterized gene target we identified that appears to be strongly regulated by HNF4A. ACY3 is only known to play an important role in deacetylating mercapturic acids in kidney proximal tubules. However, an understanding of ACY3 in the context of the beta cell is largely absent.

As for targets of HNF1A, we highlighted *GPR39* as it was previously reported to be downregulated in MODY3 hiPSC-derived endocrine cells[16]. In our studies, we confirmed that HNF1A directly binds to the *GPR39* promoter to regulate its expression in beta cells. The gene encodes a G protein-coupled receptor that has been shown before to be involved in regulating islet function[52,53], and that GPR39 agonists could be a potential therapeutic target for diabetes treatment. Several other genes that showed evidence of being bound and regulated by HNF1A include *HGD* (encoding an enzyme that regulates tyrosine and phenylalanine metabolism) and *NR5A2* (encoding an orphan nuclear receptor and transcriptional activator with

overlapping roles as HNF1A due to its involvement in tissue development, and both cholesterol and glucose metabolism[54]. Interestingly, we observed increased *NR5A2* gene expression upon knockdown of *HNF1A*, which is not known as a transcriptional repressor except potentially on itself[55], suggesting complex regulatory (direct or indirect) mechanisms at play. We also identified *PAX4*, a known endocrine marker, as a target of both HNF1A and HNF4A in our samples. *PAX4* encodes a paired box domain transcription factor that marks endocrine cells, and is known to be involved in pancreatic islet development and function[41]. It is also associated with MODY9. Thorough characterization of each of these shortlisted targets is beyond the scope here but the resource we present provide promising opportunities for further studies.

Finally, we used our ChIP-Seq pipeline to evaluate the functional impact of an established T2D risk variant in *HNF4A*, rs1800961, a low-frequency variant that has been associated with T2D risk at genome-wide significance in a multi-ancestry study[6,7]. Fine-mapping analyses had confirmed that rs1800961 is indeed the causal coding variant driving the observed T2D association signal[7], adding further credibility that the functional study of this variant will be highly relevant to gain better understanding of underlying T2D predisposition. The overlap between specific MODY genes and T2D genetic risk suggests that both conditions share common underlying genetic and molecular etiologies, and the study of the causal genes and variants may provide critical insight into diabetes mechanisms. So far, functional studies involving these missense variants have provided piecemeal information on the functional impacts of the T2D risk variants. *HNF4A* rs1800961 maps to the cytoplasmic loop hinge region of the protein. Early studies showed varying levels of LOF in terms of transactivation activity on specific target promoters but lacked consistency[56,57]. As these experiments were not conducted in relevant human beta cells, there remains a need to carry out more thorough investigations on the molecular consequences of the variants in human beta cells. What has been consistent, is that these variants do not affect protein expression levels or cellular localization, and therefore the most likely molecular impact, if any, would be on gene regulation. Our results suggested that T117I (based on HNF4A8 isoform) was able to activate several specific downstream targets such as *AKAP1*, *GAD2*, and *HOPX*. Further studies will be required to assess whether modulation of these genes affects beta cell function. Importantly, different isoforms of HNF4A not only have different tissue expression patterns but also exhibit varying transactivation activities in an isoform- and gene-specific manner[30,34,58], therefore the impact of different isoforms on the gene targets we have highlighted need to be explored more deeply to provide a holistic understanding of the diverse roles that HNF4A proteins play. Our structural modeling also demonstrated that phosphorylation of T117 could attenuate DNA binding in WT HNF4A while the T117I mutation has no detrimental impact on DNA binding and will not be subject to phosphorylation events. Carriers of the rs1800961 variant could therefore give rise to differential binding patterns of HNF4A to DNA that may influence downstream beta cell function.

In conclusion, the work presented here illustrates an approach for target discovery using ChIP-based high throughput data followed by molecular validation. Our data provides a valuable resource to investigate the downstream targets of HNF4A and HNF1A, to identify molecular mechanisms in normal beta cells and hepatic cells as well as their dysfunction in the context of diabetes. Importantly, our work paves the way for new opportunities for pre-clinical validation of potential therapeutic targets to improve beta cell function in the context of both MODY and T2D. As new data from other studies of HNF4A and HNF1A emerge, this resource will continue to be a relevant reference for gene targets in human beta cells and hepatic cells. Our approach further exemplifies a framework that can be extended to other transcriptional regulators and cell types.

## Methods

### Human pluripotent stem cell (hPSC) culture

H9 cells (WAe0009-A, WiCell) and iAGb cells (hiPSC line previously generated in our lab from human fibroblast cells from a healthy donor[59]) were cultured at 37 °C with 5% $CO_2$ in TeSR-E8 or mTeSR media (STEMCELL Technologies). The use of human cells is covered by A*STAR IRB 2020-096. HPSC media was replaced every 24 h and cells were passaged using manual hand-picking or ReLeSR (STEMCELL Technologies) weekly. All hPSC lines used were routinely tested to be mycoplasma-negative.

### Pancreatic progenitor differentiation

HPSCs were differentiated into PPs in adherent cultures as adapted from a previous protocol[14]. Briefly, cells were dissociated using ReLeSR (STEMCELL Technologies) and plated in mTeSR on plates pre-coated with 0.1% gelatin. Cells were differentiated 2 days later in RPMI-1640 (Gibco) with 2% B-27 (no vitamin A; serum-free chemically-defined medium) (Thermo Fisher Scientific) and supplemented with 100 ng/ml Activin A (R&D Systems), 3 µM CHIR99021 (Tocris) and 10 µM LY294002 (LC Labs). On day 3, differentiation medium containing 50 ng/ml Activin A was added. On day 5, differentiation medium containing 50 ng/ml FGF2 (Miltenyi Biotec), 3 µM all-trans-retinoic acid (RA) (WAKO), and 10 mM nicotinamide (Sigma Aldrich) was added, followed by subsequent media changes and addition of 20 µM DAPT (Abcam) on days 10 and 12. Cells were harvested at day 14 during the PP stage. Further details on the commercial source of reagents may be found in Table S1.

### Hepatic differentiation

HPSCs were differentiated into hepatocyte-like cells by adapting a previously published protocol[60], with some modifications[14]. The basal differentiation media used was the same as that for the PP differentiation protocol described above during the first 9 days. The same small molecules were also used for the first 4 days of differentiation. On day 5, differentiation medium containing 50 ng/ml Activin A (R&D Systems) was added. From days 6 to 8, differentiation media supplemented with 20 ng/ml BMP4 (Miltenyi Biotec) and 10 ng/ml FGF10 (Miltenyi Biotec) was added and replaced daily. From days 10 to 24, the cells were differentiated in HCM Bulletkit (Lonza) basal media supplemented with 30 ng/ml Oncostatin M (Miltenyi Biotec) and 50 ng/ml HGF (Miltenyi Biotec). The differentiation media was replaced every other day. Cells were harvested at day 8 for the hepatoblast stage. Further details on the commercial source of reagents may be found in Table S1.

### Pancreatic beta cell differentiation

H9 and/or iAGb hPSCs were dissociated using TrypLE Express (Life Technologies). $1 \times 10^6$ single cells per ml of mTeSR1 (STEMCELL Technologies Inc) containing 10 µM of Rho-Kinase Inhibitor (Y-27632) (STEMCELL Technologies Inc) were seeded into Corning® CoStar® ultra-Low attachment 6-well plates, with 3–4 million cells seeded per well. After 24 h, spheroids were formed and the culture media was replaced with differentiation media to initiate differentiation. HPSCs were differentiated sequentially into definitive endoderm, primitive gut tube, pancreatic progenitors, endocrine progenitors and finally beta-like cells in the form of cell clusters, as adapted from previously published protocols[14,61]. The basal differentiation media are made up as follows. S1 media: MCDB131 (Gibco) + 8 mM D-Glucose (Sigma Aldrich) + 2.46 g/L $NaHCO_3$ (Sigma Aldrich) + 2% FAF-BSA (Proliant) + ITS-X (Thermo Fisher Scientific) 1:50000 + 2 mM GlutaMAX (Thermo Fisher Scientific) + 0.25mM L-ascorbic acid (Sigma Aldrich) + 1% Pen/Strep (Gibco); S2 media: MCDB131 + 8 mM D-Glucose + 1.23 g/L $NaHCO_3$ + 2% FAF-BSA + ITS-X 1:50000 + 2 mM GlutaMAX + 0.25mM L-ascorbic acid + 1% Pen/Strep; S3 media: MCDB131 + 8 mM

D-Glucose + 1.23 g/L NaHCO₃ + 2% FAF-BSA + ITS-X 1:200 + 2 mM GlutaMAX + 0.25 mM L-ascorbic acid + 1% Pen/Strep; S5 media: MCDB131 + 20 mM D-Glucose + 1.754 g/L NaHCO₃ + 2% FAF-BSA + ITS-X 1:200 + 2 mM GlutaMAX + 0.25 mM L-ascorbic acid + 1% Pen/Strep + Heparin 10 μg/ml (Sigma Aldrich); S6 media: CMRL 1066 Supplemented (Mediatech) + 10% FBS (HyClone) + 1% Pen/Strep. The media changes are as follows. Day 0: S1 + 100 ng/ml Activin A (R&D Systems) + 3 μM CHIR99021 (Tocris); Day 1: S1 + 100 ng/ml Activin A; Days 3, 5: S2 + 50 ng/ml FGF7 (Miltenyi Biotec); Days 6, 7: S3 + 50 ng/ml FGF7 + 0.25 μM Sant-1 (Santa Cruz) + 2 μM RA (Wako) + 500 nM PdBU (Tocris) + 200 nM LDN193189 (Sigma Aldrich) (only on Day 7); Days 8, 10, 12: S3 + 50 ng/ml FGF7 + 0.25 μM Sant-1 + 100 nM RA; Days 13, 15: S5 + 0.25 μM Sant-1 + 100 nM RA + 1 μM XXI (Merck) + 10 μM Alk5iII (Enzo) + 1 μM T3 (Merck Millipore) + 20 ng/ml betacellulin (Cell Signaling); Days 17, 19: S5 + 25 nM RA + 1 μM XXI + 10 μM Alk5iII + 1 μM T3 + 20 ng/ml betacellulin; Days 20–35 (media change every other day): S6 + 10 μM Alk5iII + 1 μM T3. In general, one entire 6-well plate of beta cell clusters are required for a single ChIP experiment. For qPCR experiments, one well of a 6-well plate of beta cell clusters is sufficient, with biological triplicates analyzed for each cell line. Further details on the commercial source of reagents may be found in Table S1.

## Clonal cell line culture

HepG2 cells were purchased from ATCC (HB-8065) and cultured in DMEM/Low glucose (Hyclone) with 10% heat-inactivated FBS (Hyclone) and 1% NEAA (Thermo Fisher Scientific). EndoC-βH1 cells[62] were purchased from Univercell Biosolutions and cultured in DMEM/Low glucose (Gibco) supplemented with BSA (Sigma Aldrich), 2 mM GlutaMAX supplement, 50 μM 2-mercaptoethanol (Gibco), 10 mM nicotinamide (Sigma Aldrich), 5.5 μg/ml transferrin (Sigma Aldrich) and 6.7 ng/ml sodium selenite (Sigma Aldrich) on plates coated with 2 μg/ml fibronectin (Sigma Aldrich) and 1% ECM (Sigma Aldrich). Ad293 cells (STR-240085, Agilent) were cultured in DMEM/High glucose (Hyclone) with 10% heat-inactivated FBS and 1% NEAA. All cell lines used were routinely tested to be mycoplasma-negative. For cells to be used for ChIP, at least two confluent 100 mm plates were required per ChIP sample.

## Human islet processing

Human islets were obtained from the Clinical Islet Laboratory at the University of Alberta and Alberta Islet Distribution Program. The use of human cells is covered by A*STAR IRB 2020-096. Upon receipt of shipment, islets were cultured in fresh Miami Media #1A (Corning) overnight. The islets were then washed once with DPBS, left to sediment by gravity, and subsequently used for ChIP.

## Generation of cDNA overexpression constructs

The pCDH plasmids containing *HNF4A* coding sequences were cloned previously[14]. For site-directed mutagenesis, the following primers were used to generate the rs1800961 variant, through a PCR using the Phusion polymerase (Thermo Fisher Scientific): Forward primer 5′ CTTGACCTTCGAATGCTGATCCGGTCCCG 3′; Reverse primer: 5′ CGGGACCGGATCAGCATTCGAAGGTCAAG 3′. The parental strand was digested following incubation with Dpn1 (NEB). Introduced mutations were verified by DNA sequencing. The pCDH plasmids containing *HNF1A* coding sequences were cloned previously[16]. To generate constructs for FLAG-based ChIP, the pCDH vector was modified to replace the single FLAG tag with 3x FLAG tag (Bio Basic). The cDNA sequences for HNF4A2 WT, HNF4A2 T139I, HNF4A8 WT, and HNF4A2 T117I were then sub-cloned into the pCDH-3x FLAG vector and packaged into lentiviruses.

## Western blot

Cells were harvested by mechanical scraping on ice and lysed in M-PER (Thermo Scientific) in the presence of protease and phosphatase inhibitors (Sigma Aldrich). Protein lysates were quantified using the BCA Assay (Thermo Scientific), separated with sodium dodecyl sulfate polyacrylamide gel electrophoresis (SDS-PAGE) using the Mini-PROTEAN Tetra Cell system (Bio-Rad) and transferred to PVDF membranes (Bio-Rad). Primary antibodies against endogenous HNF4A protein (1:1000; H1415, R&D Systems), FLAG tag (1:1000; F1804, Sigma Aldrich) or β-actin (1:5000; 3700S, Cell Signaling) were used, followed by HRP-conjugated secondary antibodies against mouse IgG (1:10000; sc-2005, Santa Cruz) or rabbit IgG (1:10000; W4011, Promega). Chemiluminescent signals were detected using Super Signal West Dura Extended Duration substrate (Thermo Scientific) and imaged using a film developer.

## Immunofluorescence staining

EndoC-βH1 cells were seeded onto glass coverslips and fixed in 4% paraformaldehyde. Cells were washed three times in DPBS, and blocked for 1 h in 5% donkey serum (Merck Millipore, S-30, USA) in 0.1% Triton-X-100 in DPBS (DPBS-T). Cells were then incubated overnight with primary antibodies for HNF4A (H1415, R&D Systems) or HNF1A (ab96777, Abcam) at a 1:100 dilution, washed three times with DPBS, and incubated for another hour with secondary antibody at a 1:500 dilution in DPBS-T. Cells were washed again and incubated with DAPI (Sigma Aldrich, D9542-5MG, USA) for 20 min. Coverslips were mounted onto glass slides and visualized using the Olympus FV1000 Fluoview Inverted Confocal microscope.

## Flow cytometry analysis

The hPSC-derived D35 cell clusters were dissociated into single cells using TrypLE at 37 °C for 15 min and passed through a 40μm cell strainer. Single cells were fixed with 4% paraformaldehyde for 30 min, and blocked in 5% FBS in DPBS with 0.1% Triton X-100. The cells were then incubated with primary antibodies for HNF4A (H1415, R&D Systems) or HNF1A Systems (ab96777, Abcam) at a 1:100 dilution for 1 h at room temperature. Cells were washed twice and incubated with secondary antibodies for Alexa Fluor® 488 at a 1:500 dilution in the dark for 1 h at room temperature. The cells were then washed twice and finally resuspended in cold DPBS and analyzed with the BD LSR II Flow Cytometer (BD Biosciences). Data analysis was performed using the FlowJo v7.0 software.

## Chromatin immunoprecipitation (ChIP)

ChIP was performed based on a previously published protocol from our lab[21]. Briefly, hPSC-derived cells that are cultured as monolayer cultures or suspension clumps, human islets, or adherent cell lines were dissociated into single cells using TryLE. The single cell suspension was cross-linked with 3.3 mg/ml of dimethyl 3,3′-dithiobispropionimidate and 1 mg/ml of 3,3′-dithiodipropionic acid di(N-hydroxysuccinimide ester) (both Sigma Aldrich) for 15 min at room temperature, followed by 1% formaldehyde (Amresco) for 15 min at room temperature. The cross-linking reaction was quenched with 0.125 M glycine and cells were lysed to extract the nuclear fraction. Nuclei were sonicated for 30 s on/45 s off for 10 cycles using a Q500 sonicator (QSonica) with microtip probes at 30% power. Sonicated samples were pre-cleared using 10 μg rabbit or mouse IgG (Santa Cruz) and Protein A/G agarose beads. Agarose beads were removed by centrifugation and a portion of the supernatant was collected as the input control. Samples were divided equally and incubated with 10 μg of HNF4A antibody (refer to Table S1) or FLAG antibody (Sigma Aldrich, F1804), or in the case of controls, rabbit IgG or mouse IgG overnight at 4 °C. The HNF4A antibody (R&D Systems, H1415) used for the bulk of the HNF4A ChIP-Seq, binds near the C-terminal end of the protein and was previously shown to detect most of the major isoforms of HNF4A, including HNF4A1/2/4/5/7/8/10/11[34]. Further details on the commercial source of reagents may be found in Table S1. The immunoprecipitated DNA was eluted from the

beads and extracted by phenol/chloroform extraction. Finally, qPCR validation was carried out on the input, pulldown, and IgG samples using SYBR Green (BioRad), targeting the *HNF1A* or *HNF4A* promoter or a control region in *GAPDH*. For validation of the genomic regions identified in this study, primers were designed to target the bound region (spanning 150–200 bp) based on the ChIP-Seq peak sequence. Refer to Table S2 for primer sequences. QPCR data were quantitated using a standard curve based on the input DNA and normalized against *GAPDH*. Results are expressed as fold change for pulldown relative to IgG control.

## ChIP-Seq library preparation

Library preparation was done using a commercially available kit, NEB# E7645 NEBNext Ultra II DNA Library Prep Kit for Illumina KIT® following the manufacturer's protocol. During the library construction, the ChIP-ed DNA underwent End Repair, Adapter ligation, size selection (for ~150 bp insert) and 16 cycles PCR enrichment to generate the final sequencing ready library. The final library should have a fragment size of ~270 bp insert plus adapter and PCR primer sequences. The quality of the library was checked using Agilent D1000 ScreenTape. A single peak was expected and was observed indicating that the library was good and suitable for sequencing. The different libraries were then pooled together and QC using Agilent high sensitivity DNA kit and KAPA quantification.

## Cluster generation and sequencing

The samples were linearized with 0.1N NaOH into single-stranded forms, they were then neutralized and diluted into 4pM loading concentration with Hybridization buffer (HT1). The NEXTSEQ High Output was performed using the Illumina NEXTSEQ 500 Sequencers with the Illumina® Reagent v2 (75 cycle kit) Kit. The DNA were attached to the flow cell surfaces and amplified to clusters and attached with the Sequencing primers and run at $1 \times 76$ cycles, generating Single-Read 75 base-pair reads. The images were captured by the NextSeq Control Software (NCS), and the Real Time Analysis (RTA) software converted the images into Basecall (bcl) files. All the bcl files were then transferred to the server for storage and primary analysis.

## Primary analysis

In the primary analysis, the bcl files were converted into fastq files using the bcl2fastq. After the conversion, the fastq reads were filtered to remove all the reads that did not pass filtering, leaving only useable Passed Filtered (PF) reads. The primary analysis result was then generated as the Demultiplexed_Stats file and reviewed, and the PF fastq files were then passed on for further analysis.

## Peak calling and visualization

FASTQC (https://www.bioinformatics.babraham.ac.uk/projects/fastqc/) was used to get a quick pre-alignment impression of the raw sequence files. Before alignment, the adapter sequences were filtered out using BBDuk (BBMap – Bushnell B. – sourceforge.net/projects/bbmap/) from BBTools suite. The sequenced reads were aligned to genomic positions using Bowtie2[63]. The reference indexes for Bowtie2 were built using the GRCh38 assembly of the human genome[64]. Post alignment conversion from text-based SAM files to binary BAM, sorting, and removal of duplicate and unmapped reads were done using Samtools[65] and Sambamba[66]. Before peak calling, the sorted BAM files were visually explored in the Integrative Genomics Viewer (IGV)[67]. For the peak calling step, the analysis was performed twice using MACS2[68], each time with a different $q$-value threshold (default threshold $q = 0.05$ and relaxed threshold $q = 0.1$). Downstream analysis such as annotation of peaks to the nearest gene, quality assessment, summarization, and visualization of peaks coverage was performed using the R/Bioconductor packages ChIPseeker[69] and ChIPQC[70]. Additionally, using ChIPseeker, functional enrichment analysis was performed to identify enriched GO terms and KEGG pathways. Enriched GO terms and pathways were analyzed using one-sided Fisher's exact test with adjusted $p$ value cutoff of 0.05 and using the Benjamini-Hochberg multiple testing procedure. To identify enriched sequence motifs and their resemblance to any known transcription factors, the data was further analyzed using HOMER[71] motif discovery algorithm. The HNF1A ChIP-Seq data for D20 EP in this study is the same as that reported previously[16].

## Cell transfection

For gene expression and transactivation analyses, transient transfections for gene knockdown or overexpression were performed in EndoC-βH1 cells, HepG2 cells or Ad293 cells. For gene knockdown studies, 100 nM ON-TARGETplus siRNA (Horizon Discovery) was used together with 1.5 µl of Lipofectamine RNAiMAX (Thermo Fisher Scientific) per well of a 24-well plate. For *HNF4A* or *HNF1A* knockdown, the single siRNA oligonucleotide (J-003406-09 and J-008215-06 respectively) was used. For other gene knockdowns, pooled siRNAs (*ACY3*: L-010400-01; *CDKN2AIP*: L-021030-01; *HAAO*: L-008666-01; *MAP3K11*: L-003577-00; *USH1C*: L-020028-00; *VIL1*: L-012383-00) were used. The non-targeting siRNA pool (D-001810-10) was used as a control. For overexpression studies in Ad293, plasmid constructs were transfected using Lipofectamine 2000 (Thermo Fisher Scientific) at a ratio of 1:3 of DNA (in µg):Lipofectamine reagent (in µl). For overexpression in EndoC-βH1 or HepG2 cells, plasmid constructs were transfected using FuGene 6 (Promega) at a ratio of 1:6 of DNA (in µg):FuGene reagent (in µl). For transactivation studies, cells were co-transfected with the respective pGL4.10 promoter construct, pRL-TK renilla vector, and overexpression vector (Empty pCDH vector or pCDH-HNF4A WT or variant). Transfection mixes were prepared in Opti-MEM® media (Gibco) and added to cells that have been freshly seeded 1–2 days before in culture media. The culture media was replaced with fresh media 24 h after transfection. Cells were harvested 48 h post-transfection for gene expression studies, 120 h post-transfection for luciferase assays, and 96 h post-transfection for GSIS experiments. Cells were transfected in triplicate wells and each experiment was independently performed at least three times.

## Luciferase reporter assays

The following promoters/regulatory regions were cloned into the pGL4.10 vector (annotation of position with respect to start codon): *ACY3* (+4278 to −787), *CDKN2AIP* (−1841 to 152), *GPR39* (−2092 to −1), *HAAO* (−279 to +789), *MAP3K11* (−1914 to +61), *USH1C* (−197 to +2836), *VIL1* (−6009 to −3219). The *HNF1A* promoter in the pGL4.10 vector was described previously[14]. Cells were harvested 96–120 h after transfection, and luciferase activity was measured using the Dual Luciferase Assay System (Promega). Firefly luciferase activity was normalized to Renilla luciferase activity for each well to obtain relative luciferase units (RLUs). The RLUs were further normalized to the control condition within each experiment.

## RNA extraction, reverse transcription and quantitative real-time PCR

The NucleoSpin RNA Kit (MN) or TRIzol reagent was used to extract total RNA from differentiated hPSCs or cell lines respectively, according to the manufacturer's instructions. Purified RNA was reverse transcribed using the High-Capacity cDNA Reverse Transcription Kit (Applied Biosystems). QPCR was performed on the CFX384 Touch™ Real-Time PCR Detection System with iTaq™ Universal SYBR® Green Supermix (Bio-Rad). Reported fold changes are based on relative expression values calculated using the $2^{-\Delta\Delta Ct}$ method with normalization to actin expression for each sample. QPCR primers were custom-designed to span exon-exon junctions, wherever possible, using Primer-BLAST (NCBI). Sequences of primers used are listed in Tables S2 and S3.

## Glucose-stimulated insulin secretion (GSIS)

EndoC-βH1 cells were seeded at a density of $5 \times 10^5$ cells per well of a 24-well plate and transfected as described above. Cells were washed once with DPBS and incubated for an hour at 37 °C in 2.8 mmol/L glucose (Wako, 049-31165) in Krebs Ringer Buffer (KRB). Cells were washed once with KRB (without glucose) and starved for another hour in 2.8 mmol/L glucose in KRB. The supernatant was harvested to analyze baseline levels of insulin. Cells were then washed again with KRB (without glucose), followed by another hour of incubation in 16.7 mmol/L glucose in KRB. The supernatant was collected to analyze insulin levels at high glucose stimulation. Supernatants were centrifuged at $800\,g$ for 5 min at 4 °C, and samples were stored at −80 °C until further analysis. The total insulin content in the cells was harvested using acid ethanol lysis. The amount of insulin in the samples was subsequently quantified by ELISA (Mercodia, 10-1113-10), according to manufacturer's instructions. Absorbance readings were taken using the Infinite 200 plate reader (Tecan).

## Generation of HNF4A-overexpressing EndoC-βH1 stable lines

To generate EndoC-βH1 cells that are stably overexpressing empty vector, HNF4A2 WT, HNF4A2 T139I, HNF4A8 WT or HNF4A8 T117I, cells were transduced with lentiviral pCDH overexpression constructs containing the 3× FLAG tag at MOI of 20. The culture media was replaced with fresh media the next day. After 48 h transduction, cells were selected with 4 μg/ml of puromycin for 4 days, and the remaining cells were passaged and routinely maintained in growth media containing low dose of 1 μg/ml puromycin. Overexpression of the recombinant HNF4A WT or variants was confirmed by qPCR and western blot analysis.

## Statistical analysis

The statistical tests and number of independent experiments (n) conducted are indicated in the figure legends. Data represent mean ± SD unless otherwise stated. ChIP qPCR and gene expression from knockdown data were analyzed using two-tailed unpaired Student's $t$ test. Luciferase data were expressed as relative Firefly/Renilla luciferase units (RLUs) normalized to the control condition. Mean differences in normalized RLUs from the empty control were analyzed using one-way ANOVA for Ad293 experiments, while two-way ANOVA was used for EndoC-βH1 and HepG2 experiments to analyze differences segregated by knockdown and overexpression conditions. GSIS data were presented as absolute insulin concentration (μg/L), stimulation index (insulin secreted at 16.7 G/insulin secreted at 2.8 G), and secreted insulin normalized to total insulin content in the cells. These data were analyzed using one-way ANOVA. All results were considered to be significant at $p < 0.05$.

## Molecular dynamics simulations

**Preparation of structures.** The crystal structure of homodimeric HNF4A bound to its DNA response element, coactivator peptides, and myristoate (PDB code 4IQR[35]) was used as the starting structure for MD simulations. There are two copies of the complex in the PDB structure. The complex assembly that contains chains A, B, C, D, I, and J was chosen for the simulations as it generally has lower B-factors than the other complex assembly. Unresolved residues were modeled using the ModLoop web server[72] while the T117I mutation and phosphorylated T117 were introduced using PyMOL[73]. The N- and C- termini of HNF4A and the coactivator peptides were capped by acetyl and N-methyl groups, respectively. The software package PDB2PQR[74] was used to determine the protonation states of residues. Each system was solvated with TIP3P water molecules[75] in a periodic truncated octahedron box containing sodium counterions and 0.15 M NaCl, such that the box walls were at least 10 Å away from the complex.

**Molecular dynamics.** Energy minimizations and MD simulations were performed with the PMEMD module of AMBER 18[76], using the ff14SB[77] force field for the protein, the cationic dummy atom model[78] for the zinc ions in HNF4A, OL15[79] force field for the DNA and generalized AMBER force field[80] for myristoate. Atomic charges for myristoate were derived using the R.E.D. Server[81] by fitting restrained electrostatic potential (RESP) charges[82] to a molecular electrostatic potential computed by the Gaussian 16 program[83] at the HF/6-31 G* level of theory. Four independent MD simulations with different initial atomic velocities and seeds for the pseudorandom number generator were carried out on each of the systems. All bonds involving hydrogen atoms were constrained by the SHAKE algorithm[84], allowing for a time step of 2 fs. Nonbonded interactions were truncated at 9 Å, while the particle mesh Ewald method[85] was used to account for long-range electrostatic interactions under periodic boundary conditions. Weak harmonic positional restraints with a force constant of $2.0\,\text{kcal mol}^{-1}\,Å^{-2}$ were placed on the non-hydrogen atoms of the complex during the minimization and equilibration steps. Energy minimization was carried out using the steepest descent algorithm for 1000 steps, followed by the conjugate gradient algorithm for another 1000 steps. The systems were then heated gradually to 300 K over 50 ps at constant volume before equilibration at a constant pressure of 1 atm for another 50 ps. Subsequent unrestrained equilibration (2 ns) and production (500 ns) runs were carried out at 300 K and 1 atm. The Langevin thermostat[86] was used to maintain the temperature with a collision frequency of $2\,\text{ps}^{-1}$. Pressure was maintained by a Berendsen barostat[87] with a pressure relaxation time of 2 ps.

**Binding free energy calculations.** Binding free energies for the complexes were calculated using the molecular mechanics/generalized Born surface area (MM/GBSA) method[88] implemented in AMBER 18[76]. Two hundred equally-spaced snapshot structures were extracted from the last 140 ns of each of the trajectories, and their molecular mechanical energies calculated with the sander module. The polar contribution to the solvation free energy was calculated using the modified generalized Born (GB) model described by ref. [89], with the solute dielectric constant set to 4 and the exterior dielectric constant set to 80. The nonpolar contribution was estimated from the solvent accessible surface area using the linear combinations of pairwise overlaps method[90], with $\gamma = 0.005\,\text{kcal Å}^{-2}$ and $\beta = 0$. The contribution of conformational entropy was considered to be the same for all complexes and therefore ignored, as the receptors are structurally very similar and the ligand is unchanged[91].

## Reporting summary

Further information on research design is available in the Nature Portfolio Reporting Summary linked to this article.

# Data availability

All data needed to evaluate the conclusions in the paper are present in the paper and/or the Supplementary Materials. The ChIP-Seq dataset generated in this study have been deposited on GEO: GSE206240. The processed ChIP-Seq data generated in this study are provided as Data S1, S2, S3, and S6. The HepG2 ChIP-Seq data used for comparison were obtained from the ENCODE database (ENCSR000BLF and ENCSR800QIT). MD simulation input files, initial and final coordinate files have been deposited in Zenodo and are available at https://doi.org/10.5281/zenodo.10011995. The crystal structure of HNF4A used is from PDB code 4IQR. Source data are provided with this paper.

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

## Acknowledgements

We thank members of the Teo lab for technical assistance, in particular Resilind Su Ern Chew, Elhadi Iich, Shirley Suet Lee Ding, Duong Tien Quang Huy and Dominique Mah. We also thank the Teo lab members for critical reading and feedback of this manuscript. We thank the Genome Institute of Singapore (GIS) Integrated Genome Analytics Platform, A*STAR, for sequencing of the ChIP DNA samples. We further thank Tatsuya Kin and James Shapiro of the Alberta Islet Distribution Program for the human islet samples. Some figures were created with BioRender.com. N.H.J.N. is supported by the A*STAR Graduate Academy, NMRC OFYIRG18may040 and the first A*STAR Career Development Fund 2019 202D800020. M.C.M. is supported by A*STAR's Singapore International Pre-Graduate Award (SIPGA). Y.S.T. is supported by BII, A*STAR and the National Medical Research Council (OFYIRG21nov-0039). A.K.K.T. is supported by IMCB, A*STAR, FY2019 SingHealth Duke-NUS Surgery Academic Clinical Programme Research Support Programme Grant, Precision Medicine and Personalised Therapeutics Joint Research Grant 2019, the 2nd A*STAR-AMED Joint Grant Call 192B9002, HLTRP/2022/NUS-IMCB-02, Paris-NUS 2021-06-R/UP-NUS (ANR-18-IDEX-0001), OFIRG21jun-0097, CSASI21jun-0006, MTCIRG21-0071, SDDC/FY2021/EX/93-A147, FY 2022 Interstellar Initiative Beyond grant, H22G0a0005, HLCA23Feb-0031, SC36/19-000801-A044 and H24G1a0015.

## Author contributions

Conceptualization: N.H.J.N. and A.K.K.T.; Methodology: N.H.J.N., S.G., S.H., Y.S.T. and A.K.K.T.; Formal Analysis: N.H.J.N., S.G., M.C.M., Y.S.T. and S.H.; Investigation: N.H.J.N., S.G., C.M.B., C.C., B.S.J.L., J.T.C., M.C.M., Y.S.T. and E.L.; Writing—original draft: N.H.J.N.; Writing—review & editing: N.H.J.N., Y.S.T., and A.K.K.T.; Funding acquisition: A.K.K.T.

## Competing interests

N.H.J.N. and A.K.K.T. are co-founders and shareholders of BetaLife Pte Ltd but are not employed by BetaLife Pte Ltd. The remaining authors declare no competing interests.
