## [Peer Review File · Nature Communications]

HNF4A and HNF1A exhibit tissue specific target gene regulation in pancreatic beta cells and hepatocytesReviewers' comments:

Reviewer #1 (Remarks to the Author):

In this manuscript, the authors study HNF1A and HNF4A using the human pluripotent stem cell model. They perform ChIP-seq for both HNF1A and HNF4A in stem cell derived pancreatic cells as well as cell lines and primary islets. The authors also examine ChIP of an HNF4A variant associated with T2D. They then go on to validate some of the HNF4A targets, by siHNF4A treatment and gene expression. They find that a MODY1 HNF4A variant was less able to rescue expression of several target genes in promoter assays. HNF4A targets HAAO and USH1C were found to impact insulin secretion in endoC-Bh1 cells. Similar experiments were performed in HEPG2 cells to study HNF4A targets in hepatocytes. HNF1A targets were also validated in KD studies but no functional validation was performed. The authors examine genes that are co-occupied by both HNF1A and HNF4A in beta cells. Lastly, the authors examine a T2D HNF4A variant and find a few sites it may bind more efficiently. Overall, this manuscript provides useful ChIP-seq datasets and describes potential new targets of HNF4A and HNF1A that could regulate beta cell and hepatocyte function. With that being said, the manuscript was a little hard to follow and the overall impact of these studies are somewhat incremental and may be better suited to a more specialty journal. While I give comments to help improve the manuscript, even if addressed I still think this would be better suited to another journal unless a much more detailed analysis is performed with greater insights into HNF4A and/or HNF1A biology. See below for specific concerns.

Concerns/Questions:

- 1) Some of the most interesting data from this study are the impacts of HAAO and possibly USH1C in beta cell function using the endoc-bh1 cell line. These studies are preliminary in nature and the differences seem small. Further studies validating these genes as important in beta cell function would be helpful. This could entail studying these in stem cell models, primary islets, and/or more detailed studies in endoc cells.
- 2) While beta cell targets for HNF1A were validated in a beta cell line, no functional validation was performed. This would be helpful in determining if these targets regulate beta cell function.
- 3) In figure 7, the authors go on to further characterize the T2D HNF4A variant, focusing on the A8 version as opposed to the A2 version. It is not clear why they choose the smaller variant when in earlier studies in figures 3 and 4 shows the A8 variant has weak transactivation activity. Are these results confirmed with the A2 variant? The results in Fig 7H, showing differences of the A8 isoform of the T2D variant on target gene expression is somewhat small and may be larger with the more active isoform.

Minor concern:

- 1) The authors in figure 3 and figure 4 compare the A2 and A8 isoforms of HNF4A. The way the data is presented can be somewhat confusing to the reader. Since the A8 isoform is known to have poor transactivation activity which is confirmed here, it might be better to include only data from the A2 isoform for clarity purposes and to include A8 data in supplemental figures.

Reviewer #2 (Remarks to the Author):

This study addresses an important topic – direct targets of the transcription factors HNF4a and HNF1a in human beta cells, which have not been well characterized. ChIPseq on HNF4a has been performed in HepG2 (human liver cancer) cells by several groups but not in combination with HNF1a. The hPSC model with derivation of the pancreatic and hepatic lineages is an important

advancement. The authors also perform follow up experiments to verify the genes as direct targets (and follow up with effects of HNF1/4 targets on insulin secretion). They compare different HNF4a isoforms and analyze an HNF4a coding variant (T117I) that has not been well studied, and identify several genes upregulated by this variant which are of interest.

Overall the work is very well done and nicely presented although there were a couple important omissions that are noted below. In short, new target genes for HNF4a and HNF1a are identified in human beta cells and hepatocytes and this will be a valuable resource for the field. The authors provide a thorough and appropriate description of the limitations of the study. The major critique of this work is that the main issue of the role of HNF4a and HNF1a, two MODY genes, in insulin secretion in beta cells is not really answered, although new target genes were identified. Synergy other than joint ChIPseq peaks was not investigated and no new mechanisms nor new concepts were presented (other than some new target genes).

Major Concerns:

It is not clear which HNF4a isoform(s) the Cell Signaling antibody detects – the reader should not have to look this up on the company website. It is also important when different antibodies are used in different ChIPseq experiments.

HepG2 is a cancer cell line while the beta cells are PSC-derived. The authors should discuss how these differences might impact their results.

Fig 3B and 3C – in general not a big effect on gene expression of targets by introduction of the siHNF4A or the cDNA for HNF4a WT or variants.

Fig 3D – isolated promoter constructs – better effect of HNF4a2 WT vs HNF4a8 small or no effect – it is not clear why the authors are they also using siHNF4a in this experiment.

Is it known which HNF4a isoforms is in their various cell types – HNF4a2 or HNF4a8? Others have shown quite some time ago that the HNF4a2 and a8 isoforms have different transactivation activity and recruit different cofactors. The authors see a similar general trend here but should consider citing some previous work by Torre-Padilla and Weiss.

Potential subtle alterations in transcriptional activity of MODY mutants is a very interesting topic especially if it involves changes in DNA binding specificity which could result in differences in target genes. But it is not completely clear why the authors chose the HNF4a variant T117I for analysis. Presumably T117 is in the DNA binding domain (the authors should indicate exactly where). Also the exact differences in binding specificity between T117I and WT are not clear – given that the DNA binding motifs for HNF4a in Fig 1E and Fig 7D are the reverse complement of each other made that comparison even more difficult.

It was nice that common targets for HNF4A were identified and liver and pancreatic cells but target genes specific to a given cell type would also be of interest and raise interesting questions about mechanism.

In Table S1 and S2 and Fig 6C common ChIPseq targets to HNF4A and HNF1A are identified – it might have been useful to follow some of those up to see if there was any transcriptional synergy between these two factors, especially given that they are both MODY genes. For example, they could employ the siRNAs – eg siHNF1A when testing HNF4a constructs. It would have been helpful to identify (by eye if necessary) potential binding sites in the ChIPseq peaks in the T117I-unique target genes.

Fig 4C, 5C – no scale given for ChIPseq peaks – is the same scale used for the same gene across different cell types? How do they compare between different target genes?

Fig 1, 4, 5 etc -- No chromosomal location numbers are given or the regions are too large to be informative (Fig 1F) making it difficult for others to follow up. While it is understood that all the

data are available in the original ChIPseq files and that easy-to-read visualizations are needed for the first level of review/understanding, some indication of the chromosomal location number in these figures would allow others to follow up on specific target genes, which is one of the stated goals of this work.

Fig 7C and other transfections – need to normalize to amount of HNF4a protein expressed – important for variants as well as the different isoforms, especially for the ChIPseq experiments

Minor:

Fig 1E – what is the source of the consensus motifs? Fewer peaks in pancreatic cells could be due to the fact that they tend to have more proteases and nucleases than hepatic cells.

Cite original reference for HNF4A/1A binding each other promoters – did they examine the same site?

Several places where Syntax or word choice needs to be fixed throughout.

Point-by-point response to reviewers' comments

Reviewers' comments:

Reviewer #1 (Remarks to the Author):

In this manuscript, the authors study HNF1A and HNF4A using the human pluripotent stem cell model. They perform ChIP-seq for both HNF1A and HNF4A in stem cell derived pancreatic cells as well as cell lines and primary islets. The authors also examine ChIP of an HNF4A variant associated with T2D. They then go on to validate some of the HNF4A targets, by siHNF4A treatment and gene expression. They find that a MODY1 HNF4A variant was less able to rescue expression of several target genes in promoter assays. HNF4A targets HAAO and USH1C were found to **impact insulin secretion** in endoC-Bh1 cells. Similar experiments were performed in HEPG2 cells to study HNF4A targets in hepatocytes. HNF1A targets were also validated in KD studies but no functional validation was performed.

- In revised Figure 3F, we do demonstrate functional validation in EndoC-bH1 cells. R#1 mentioned that we validated HNF1A targets in revised Figure 5. We can certainly perform similar functional validation experiments when given the opportunity to resubmit our manuscript.

The authors examine genes that are co-occupied by both NF1A and HNF4A in beta cells. Lastly, the authors examine a T2D HNF4A variant and find a few sites it may bind more efficiently. Overall, this manuscript provides useful ChIP-seq datasets and describes potential new targets of HNF4A and HNF1A that could regulate beta cell and hepatocyte function.

- We thank R#1 for acknowledging that we provide useful ChIP-Seq datasets and describe potential new targets of HNF4A and HNF1A that could regulate beta cell and hepatocyte function.

With that being said, the manuscript was a little hard to follow and the overall impact of these studies are somewhat incremental and may be better suited to a more specialty journal. While I give comments to help improve the manuscript, even if addressed I still think this would be better suited to another journal unless a much more detailed analysis is performed with greater insights into HNF4A and/or HNF1A biology. See below for specific concerns.

- Following R#1 comments that the manuscript was a little hard to follow, we have revised it for the moment to provide greater clarity.
- Importantly, relating to R#1's comments that our study did not go into greater depth, we respectfully respond that we **intended this manuscript to be a resource article** instead of an original article. In this regard, we have provided a lot of data, identifying many exciting and yet-unknown targets to be followed up by the research community.
- Following discussions with the editor, she also **acknowledged and agreed that our dataset is quite of importance** and that **our manuscript has a lot of value as a Resource**.
- That said, we would be happy to **delve deeper into selected HNF4A and/or HNF1A targets to demonstrate their roles in insulin secretion** if allowed to resubmit our manuscript. Thank you.

Concerns/Questions:

1) Some of the most interesting data from this study are the impacts of HAAO and possibly USH1C in beta cell function using the endoc-bh1 cell line. These studies are preliminary in nature and the differences seem small. Further studies validating these genes as important in beta cell function would be helpful. This could entail studying these in stem cell models, primary islets, and/or more detailed studies in endoc cells.

- We will be able to perform siRNA-mediated knockdown studies to evaluate the effects of loss of HAAO and USH1C on beta cell function in greater detail in EndoC-bH1 cells.
- As the current seven main and seven supplementary figures are already very congested, in-depth mechanistic study of one or two genes in various beta cell models could be better reserved for a follow-on Original Article instead of this Resource Article. We thank you for your kind understanding.

2) While beta cell targets for HNF1A were validated in a beta cell line, no functional validation was performed. This would be helpful in determining if these targets regulate beta cell function.

- As mentioned above, we will be able to perform siRNA-mediated knockdown studies to evaluate the effects of loss of some targets in Figure 5. We would also be happy to delve deeper into a HNF1A target (e.g. GPR39) to demonstrate its role in insulin secretion if allowed to resubmit our manuscript. Thank you.

3) In figure 7, the authors go on to further characterize the T2D HNF4A variant, focusing of the A8 version as opposed to the A2 version. It is not clear why they choose the smaller variant when in earlier studies in figures 3 and 4 shows the A8 variant has weak transactivation activity. Are these results confirmed with the A2 variant? The results in Fig 7H, showing differences of the A8 isoform of the T2D variant on target gene expression is somewhat small and may be larger with the more active isoform.

- We thank R#1 for this comment that HNF4A8 and HNF4A2 data could be presented better. We have hence revised the manuscript and only displayed data from one isoform at a time for clarity, particularly in Figure 3 for the transient HNF4A overexpression experiments. Our focus on the A8 isoform in Figure 7 has been clarified in the text – ChIP-Seq data derived from our stable EndoC-bH1 cell line overexpressing the A8 isoform yielded more meaningful results.

Minor concern:

1) The authors in figure 3 and figure 4 compare the A2 and A8 isoforms of HNF4A. The way the data is presented can be somewhat confusing to the reader. Since the A8 isoform is known to have poor transactivation activity which is confirmed here, it might be better to include only data from the A2 isoform for clarity purposes and to include A8 data in supplemental figures.

- We thank R#1 for this comment that HNF4A8 and HNF4A2 data could be presented better. We have hence revised the manuscript and only displayed data from one isoform at a time for clarity, particularly in Figure 3 for the transient HNF4A overexpression experiments.

Reviewer #2 (Remarks to the Author):

This study addresses an important topic – direct targets of the transcription factors HNF4a and HNF1a in human beta cells, which have not been well characterized. ChIPseq on HNF4a has been performed in HepG2 (human liver cancer) cells by several groups but not in combination with HNF1a. The hPSC model with derivation of the pancreatic and hepatic lineages is an important advancement. The authors also perform follow up experiments to verify the genes as direct targets (and follow up with effects of HNF1/4 targets on insulin secretion). They compare different HNF4a isoforms and analyze an HNF4a coding variant (T117I) that has not been well studied, and identify several genes upregulated by this variant which are of interest.

Overall the work is very well done and nicely presented although there were a couple important omissions that are noted below. In short, new target genes for HNF4a and HNF1a are identified in human beta cells and hepatocytes and this will be a valuable resource for the field. The authors provide a thorough and appropriate description of the limitations of the study.

- We thank R#2 for acknowledging that our study addresses an important point – direct targets of HNF4A and HNF1A in human beta cells, which have not been well characterized. This is also what the editor acknowledged, finding value of this important resource.
- R#2 also acknowledged that our model is an important advancement, and that the work is very well done and nicely presented, and that this is a valuable resource for the field.

The major critique of this work is that the main issue of the role of HNF4a and HNF1a, two MODY genes, in insulin secretion in beta cells is not really answered, although new target genes were identified. Synergy other than joint ChIPseq peaks was not investigated and no new mechanisms nor new concepts were presented (other than some new target genes).

- In response to R#2, we respectfully point out that we intended this to be a Resource Article and not an Original Article. Therefore, we have not delved deeply into the genes, given that the seven main and seven supplementary figures are already very congested. That said, in revised Figure 3, we did evaluate the effects of loss of HNF4A targets on insulin secretion in beta cells. As in our response to R#1 above, we will be able to perform siRNA-mediated knockdown studies to evaluate the effects of loss of some targets in Figure 5. We

would also be happy to delve deeper into the selected HNF4A targets and a HNF1A target (e.g. GPR39) to demonstrate their roles in insulin secretion if allowed to resubmit our manuscript. Thank you.

- We can also evaluate better synergy of the data and present new concepts with greater clarity. That said, we respectfully point out that we intended this to be a Resource Article and not an Original Article. As the current seven main and seven supplementary figures are already very congested, in-depth study of one or two genes in various beta cell models with new mechanistic insights could be better reserved for a follow-on Original Article instead of this Resource Article. We thank you for your kind understanding.

Major Concerns:

It is not clear which HNF4a isoform(s) the Cell Signaling antibody detects – the reader should not have to look this up on the company website. It is also important when different antibodies are used in different ChIPseq experiments.

- We are utilizing the HNF4A antibody from R&D Systems that has previously been shown to detect most major HNF4A isoforms, we have now made this clearer in the methods and in the figure legends. We thank R#2 for this comment.

HepG2 is a cancer cell line while the beta cells are PSC-derived. The authors should discuss how these differences might impact their results.

- We can discuss this point further and include it under discussion and/or limitations of the study.

Fig 3B and 3C – in general not a big effect on gene expression of targets by introduction of the siHNF4A or the cDNA for HNF4a WT or variants.

- It is known that EndoC-bH1 cells are notoriously difficult to transfect. Knockdown of *HNF4A* decreased its transcript by 50 %, correspondingly decreasing the transcript expression of its downstream targets by 50 %, such as that for *ACY3*, *USH1C* and *VIL1*. (Figure 3B). By luciferase, WT HNF4A appeared to have the strongest effect on *ACY3* transcriptional activity (Figure 3C).

Fig 3D – isolated promoter constructs – better effect of HNF4a2 WT vs HNF4a8 small or no effect – it is not clear why the authors are they also using siHNF4a in this experiment.

- Following the comment from R#1 that HNF4A8 and HNF4A2 data could be presented better, we have since revised the manuscript and only displayed data from one isoform at a time for clarity in this figure.
- siHNF4A was performed to first decrease the basal expression of endogenous *HNF4A*. This would thus allow the effects of overexpression of HNF4A on the promoter activities to be clearer.

Is it known which HNF4a isoforms is in their various cell types – HNF4a2 or HNF4a8? Others have shown quite some time ago that the HNF4a2 and a8 isoforms have different transactivation activity and recruit different cofactors. The authors see a similar general trend here but should consider citing some previous work by Torre-Padilla and Weiss.

- Torres-Padilla and Weiss is cited in reference 58. We thank R#2 for the comment.

Potential subtle alterations in transcriptional activity of MODY mutants is a very interesting topic especially if it involves changes in DNA binding specificity which could result in differences in target genes. But it is not completely clear why the authors chose the HNF4a variant T117I for analysis. Presumably T117 is in the DNA binding domain (the authors should indicate exactly where). Also the exact differences in binding specificity between T117I and WT are not clear – given that the DNA binding motifs for HNF4a in Fig 1E and Fig 7D are the reverse complement of each other made that comparison even more difficult.

- We thank R#2 for noting that we are trying to address an important topic. We have now included new structural modelling data of the T2D risk variant in HNF4A in revised Figure 7 (not previously shown before), to show that the presence of the T2D variant influences DNA binding due to differences in phosphorylation at the residue (new concept). This could explain the potential upregulation of the HNF4A protein binding to DNA sequences.

This supports our discovery of increased target gene expression found in HNF4A T2D variant-expressing cells.

It was nice that common targets for HNF4A were identified and liver and pancreatic cells but target genes specific to a given cell type would also be of interest and raise interesting questions about mechanism.

- In response to R#2, we respectfully point out that we intended this to be a Resource Article and not an Original Article. Therefore, we have not delved deeply into the genes, given that the seven main and seven supplementary figures are already very congested. That said, in revised Figure 3, we did evaluate the effects of loss of HNF4A targets on insulin secretion in beta cells. As in our response to R#1 above, we will be able to perform siRNA-mediated knockdown studies to evaluate the effects of loss of some targets in Figure 5. We would also be happy to delve deeper into selected HNF4A targets and a HNF1A target (e.g. GPR39) to demonstrate their roles in insulin secretion if allowed to resubmit our manuscript. Thank you.

In Table S1 and S2 and Fig 6C common ChIPseq targets to HNF4A and HNF1A are identified – it might have been useful to follow some of those up to see if there was any transcriptional synergy between these two factors, especially given that they are both MODY genes. For example, they could employ the siRNAs – eg siHNF1A when testing HNF4a constructs. It would have been helpful to identify (by eye if necessary) potential binding sites in the ChIPseq peaks in the T117I-unique target genes.

- We can follow up on these common ChIP-Seq targets in HepG2, using siRNAs as suggested, if allowed to resubmit our manuscript.
- Potential binding sites in the ChIP-Seq peaks for T117I have now been provided in revised Figure 7.

Fig 4C, 5C – no scale given for ChIPseq peaks – is the same scale used for the same gene across different cell types? How do they compare between different target genes?

- We have now included the scales for all ChIP-Seq peaks in Figures 3A, 4C, 5C, 7F and S7D, to provide better appreciation of how the peak enrichment compares between different target genes. Thank you.

Fig 1, 4, 5 etc -- No chromosomal location numbers are given or the regions are too large to be informative (Fig 1F) making it difficult for others to follow up. While it is understood that all the data are available in the original ChIPseq files and that easy-to-read visualizes are needed for the first level of review/understanding, some indication of the chromosomal location number in these figures would allow others to follow up on specific target genes, which is one of the stated goals of this work.

- We have now included the the chromosomal location numbers for all ChIP-Seq peaks in Figures 3A, 4C, 5C, 7F and S7D, for easier follow-up of gene regions in addition to the detailed information found in the Supplementary tables. Thank you.

Fig 7C and other transfections – need to normalize to amount of HNF4a protein expressed – important for variants as well as the different isoforms, especially for the ChIPseq experiments

- For transfections, we normalized against cell number. We also transfected the same amount of DNA into the cells. For ChIP-Seq experiments, we normalised against the qPCR for a non-targeting control region. We opine that these are appropriate methods for normalization of data.

Minor:

Fig 1E – what is the source of the consensus motifs? Fewer peaks in pancreatic cells could be due to the fact that they tend to have more proteases and nucleases than hepatic cells.

- As stated in the methods, we used HOMER motif discovery algorithm (Page 35 of 66).

Cite original reference for HNF4A/1A binding each other promoters – did they examine the same site?

- Reference 19 has been cited (Page 40 or 66). As mentioned before, the prior study was a ChIP-on-chip study while ours is a ChIP-Seq study with greater resolution. Thank you.

Several places where Syntax or word choice needs to be fixed throughout.

- We thank R#2 for this comment. We have tried to fix the syntax in this revised manuscript. We will take a closer look again if allowed to resubmit our manuscript. Thank you.

REVIEWERS' COMMENTS

Reviewer #1 (Remarks to the Author):

I agree that as a resource article this work represents important datasets that can be followed up by the scientific community.

One area that could use a little clarification are the datasets derived from human islets and D35BLCs. In both cases I'm sure that less than half of the cells are actually beta cells so the signal may be from other endocrine cells for the islets samples and from some progenitor populations in the D35BLCs. Do the authors have data demonstrating what proportion of the D35BLCs are actually insulin+ cells? This can vary greatly depending on lab. This would be helpful in interpreting the datasets. This point should also be discussed in the manuscript though it is mitigated to a certain degree by use of endoC beta cell line which is homogeneous.

Reviewer #2 (Remarks to the Author):

The authors have adequately addressed my previous concerns.

The Discussion could be tightened up a bit. Below are just a few suggestions:

l. 539 "home" should be "hone"

l. 549 "On a similar thread," can be deleted

l. 552 "When compared with trends observed in the hepatic cells, it is encouraging to note that" could be shortened to "Notably,"

l. 572- 576, discussion of USH1C can be tightened up

l. 621- l. 622

l. 645 "Beyond the results that we have presented here, the data that we shared may be mined further for additional candidate genes" can probably be deleted

l. 648 "Our study also provides a framework

that can be applied to the study of other transcriptional regulators and cell types, be it in the context of tissue development or disease modelling." can be shortened and perhaps combined with the previous sentence?

Point by point response Ng et al

Reviewer #1 (Remarks to the Author):

I agree that as a resource article this work represents important datasets that can be followed up by the scientific community.

One area that could use a little clarification are the datasets derived from human islets and D35BLCs. In both cases I'm sure that less than half of the cells are actually beta cells so the signal may be from other endocrine cells for the islets samples and from some progenitor populations in the D35BLCs. Do the authors have data demonstrating what proportion of the D35BLCs are actually insulin+ cells? This can vary greatly depending on lab. This would be helpful in interpreting the datasets. This point should also be discussed in the manuscript though it is mitigated to a certain degree by use of endoC beta cell line which is homogeneous.

Author's Response

We thank the reviewer for the suggestion. The differentiation protocol utilised in the study has been published in earlier papers from the lab (Lau et al., 2023, Low et al., 2021), which showed that the D35 β LCs possess ~35% insulin-positive cells based on flow cytometry data, with a range of 20% to 60% across multiple independent experiments and wild type hPSC lines. The cell lines used in this study are the same as that in Low et al., 2021.

We acknowledge the importance of discussing this point and therefore added the following sentences in the discussion: 'A limitation of our hPSC-based models however relates to the heterogeneity of differentiated cells. For instance, the β cell differentiation protocol used in this study generates ~35% insulin-positive cells based on previously published flow cytometry data^{16, 41}, indicating the presence of other non- β LC populations. As ChIP-Seq was conducted on bulk samples, peak signals from the D35 β LC samples could be derived from non-insulin-expressing cells as well. To address this, we overlapped the hPSC-derived cell data with our EndoC- β H1 (a homogenous human β cell line) and human islet datasets to focus only on gene targets that are common. This increases the confidence that the genes highlighted are of relevance to β cells.'

Reviewer #2 (Remarks to the Author):

The authors have adequately addressed my previous concerns.

Author's Response

The Discussion could be tightened up a bit. Below are just a few suggestions:

l. 539 "home" should be "hone" **Done.**

l. 549 "On a similar thread," can be deleted **Done.**

l. 552 "When compared with trends observed in the hepatic cells, it is encouraging to note that" could be shortened to "Notably," **Done.**

l. 572- 576, discussion of USH1C can be tightened up **Done.**

l. 621- l. 622 **It is not clear what the suggested changes are here.**

I. 645 "Beyond the results that we have presented here, the data that we shared may be mined further for additional candidate genes" can probably be deleted **Done**.

I. 648 "Our study also provides a framework that can be applied to the study of other transcriptional regulators and cell types, be it in the context of tissue development or disease modelling." can be shortened and perhaps combined with the previous sentence? **Done**.

We thank the reviewer for the suggestions. We have addressed all the edits above as well as edited the Discussion further.